# BLACK BOXES AND LOOKING GLASSES: MULTILEVEL SYMMETRIES, REFLECTION PLANES, AND CONVEX OPTIMIZATION IN DEEP NETWORKS

## ABSTRACT

We show that training deep neural networks (DNNs) with absolute value activation and arbitrary input dimension can be formulated as equivalent convex Lasso problems with novel features expressed using geometric algebra. This formulation reveals geometric structures encoding symmetry in neural networks. Using the equivalent Lasso form of DNNs, we formally prove a fundamental distinction between deep and shallow networks: deep networks inherently favor symmetric structures in their fitted functions, with greater depth enabling multilevel symmetries, i.e., symmetries within symmetries. Moreover, Lasso features represent distances to hyperplanes that are reflected across training points. These reflection hyperplanes are spanned by training data and are orthogonal to optimal weight vectors. Numerical experiments support theory and demonstrate theoretically predicted features when training networks using embeddings generated by Large Language Models.

## 1 INTRODUCTION

Recent advancements have demonstrated that deep neural networks are powerful models that can perform tasks including natural language processing, synthetic data and image generation, classification, and regression. However, research literature still lacks in intuitively understanding why deep networks are so powerful: what they "look for" in data, or in other words, how each layer extracts features. We are interested in the following question:

> *Is there a fundamental difference in the nature of functions learned by deep networks, as opposed to shallow networks?*

We answer this question by transforming non-convex training problems into convex formulations and analyzing their structure. **We show that deep nets favor symmetric structures, with greater depth representing multilevel symmetries encoded via reflections of reflections of data points.**

A *neural network* of depth $L$ is a parameterized function $f_L(\cdot; \theta) : \mathbb{R}^{1 \times d} \to \mathbb{R}$ of the form

$$f_L(\mathbf{x}; \theta) = \sum_{i=1}^{m_L} \sigma\left(\cdots\left(\sigma\left(\sigma\left(\mathbf{x}\mathbf{W}^{(i,1)} + \mathbf{b}^{(i,1)}\right)\mathbf{W}^{(i,2)} + \mathbf{b}^{(i,2)}\right)\cdots\right)\mathbf{W}^{(i,L-1)} + \mathbf{b}^{(i,L-1)}\right)\alpha_i + \xi. \tag{1}$$

The trainable parameters are the outermost weights $\alpha_i \in \mathbb{R}$, an external bias $\xi \in \mathbb{R}$; and the inner weights $\mathbf{W}^{(i,l)} \in \mathbb{R}^{m'_{l-1} \times m'_l}$ and inner biases $\mathbf{b}^{(i,l)} \in \mathbb{R}^{1 \times m'_l}$ for each layer $l$. There are $m_l = m_L m'_l$ neurons in each layer. The *activation* function is $\sigma : \mathbb{R} \to \mathbb{R}$. Note that $m'_0 = d$, which is the input data dimension, and $m'_{L-1} = 1$. Let $N$ be the number of training samples and $\mathbf{X} \in \mathbb{R}^{N \times d}$ be the *training data matrix*, where each row of $\mathbf{X}$ is a training sample $\mathbf{x}_n \in \mathbb{R}^{1 \times d}$. The neural network extends to matrix inputs row-wise. We consider regression problems, although the results can also be applied to classification problems. The *target* or *label* vector is $\mathbf{y} \in \mathbb{R}^N$. The neural network *training problem* is

$$\min_{\theta \in \Theta} \frac{1}{2}\|f_L(\mathbf{X}; \theta) - \mathbf{y}\|_2^2 + \frac{\beta}{L}\sum_{i=1}^{m}\left(|\alpha_i|^L + \sum_{l=1}^{L-1}\|\mathbf{W}^{(i,l)}\|_1^L\right) \tag{2}$$

where $\theta = \{\mathbf{W}^{(i,l)}, \mathbf{b}^{(i,l)}, \alpha_i, \xi : i \in [m_L], l \in [L-1]\}$ denotes the set of trainable parameters and $\Theta$ is the parameter space. The *regularization parameter* is $\beta > 0$, which is fixed. The $l_1$ regularization penalty in (2) is used for analytical tractability and it is known that $l_1$ regularization can approximate $l_2$ regularization Pilanci (2023a). Our results generalize to other convex loss functions in addition to $l_2$ loss, and also applies to the case when $\beta \to 0$ by using the minimum norm problem (14). Neural networks (1) and their training problems (2) are non-convex, which can make understanding and training even shallow neural networks challenging. We provide a foundational analysis that isolates the effect of neural network depth by focusing on the deep narrow network. The *deep narrow network* is a special case of (1) with absolute value activation $\sigma(x) = |x|$, arbitrary depth $L$, and a width $m_1 = \cdots = m_L = m$ that is arbitrary but constant across layers:

$$f_L(\mathbf{x}; \theta) = \sum_{i=1}^{m} \left| \cdots \left| \left| \left| \mathbf{x}\mathbf{W}^{(i,1)} + b^{(i,1)} \right| w^{(i,2)} + b^{(i,2)} \right| \cdots \right| w^{(i,L-1)} + b^{(i,L-1)} \right| \alpha_i + \xi. \quad (3)$$

Except for $\mathbf{W}^{(i,1)} \in \mathbb{R}^d$, all trainable parameters in the deep narrow network are scalars, as denoted by the lowercase, non-boldface text in (3). The total number of neurons across each of the $L$ layers is $m$. We show that increasing the number of layers in a deep narrow network even with the constant width $m_1 = \cdots = m_{L-1} = m$ enables the network to learn richer features.

The deep narrow network uses absolute value activation instead of the more standard ReLU. The absolute value activation is a symmetric function and this allows for tractable, simpler formulations of neural network features. The absolute value activation is equivalent to the ReLU activation in 2-layer networks with skip connections (Zeger et al., 2024). Moreover, neural networks with absolute value activation have been shown to perform comparably to ReLU and have advantages such as no exploding or vanishing gradients, making them suitable for deep networks (I.Berngardt, 2023) and outperforming other activations in computer vision (Vallés-Pérez et al., 2023). Our results suggest that absolute value activations are especially primed to capture reflections and symmetries in the data, leading to their representation power. Extensions to ReLU networks are discussed in Section 4.1.

Recently, (Pilanci, 2023b) described neural network features using geometric algebra, also known as Clifford algebra. Geometric algebra is an extension of vector algebra that centers around geometric operations such as translation and reflections (Perwass et al., 2009; Berger, 2009; Chisolm, 2012). Geometric algebra provides a unified framework for expressing laws of physics (Doran & Lasenby, 2003; Hestenes, 2003) and has applications in computer graphics (Vince, 2008). Geometric algebra generalizes vector operations such as cross products, inner products, and outer products from a geometric perspective Berger (2009). The use of geometric algebra, which is well-studied, also opens doors to potentially deep connections or applications to physics. In this work, we extend (Pilanci, 2023b) and leverage geometric algebra to examine the connection between deep neural networks and Lasso problems. A *Lasso problem* is a convex optimization problem of the form

$$\min_{\mathbf{z}, \xi} \frac{1}{2} \|\mathbf{A}\mathbf{z} + \xi \mathbf{1} - \mathbf{y}\|_2^2 + \beta \|\mathbf{z}\|_1 \quad (4)$$

where $\mathbf{A}$ is a *dictionary matrix*, and its columns $\mathbf{A}_i \in \mathbb{R}^N$ are *feature vectors*. We will show that the Lasso problem and the deep narrow network training problem are equivalent (Theorem 4.4). The Lasso problem, its solution set, and solvers for it are well-studied (Efron et al., 2004; Tibshirani, 1996; 2013). Thus the Lasso model sheds new light on the training and geometric interpretability of neural networks. For training, the convexity of the Lasso problem implies that all of its stationary points are optimal, which prevents convergence to sub-optimal local minima that can occur with the non-convex training problem (2). Moreover, there are efficient and interpretable algorithms such as Least Angle Regression (LARS) for solving the Lasso problem (Efron et al., 2004).

In terms of geometric interpretability, the $l_1$ regularization in the Lasso problem (4) favors a sparse solution for $\mathbf{z}$. The dictionary columns $\mathbf{A}_i$ where $z_i \neq 0$ constitute a sparse set of feature vectors that we will show correspond to *feature* functions defining an optimal neural network. We will also show that the features can be expressed using volumes and wedge products defined in geometric algebra, and that they measure distances to *reflection planes* spanned by subsets of training data. For $\mathbf{a}, \mathbf{b} \in \mathbb{R}^d$, the *reflection of* $\mathbf{a}$ *about* $\mathbf{b}$ is $R_{(\mathbf{a},\mathbf{b})} = 2\mathbf{b} - \mathbf{a}$. With respect to $\mathbf{b}$, $\mathbf{a}$ and $R_{(\mathbf{a},\mathbf{b})}$ have a *single-level symmetry*. Higher-order reflections can be defined to create deeper levels of symmetry (Section 4.2). Neural networks encode reflection planes that are based at training data and their reflections about each other, and which are orthogonal to neuron weights.

**Significance:** The Lasso model elucidates a quantitative difference between shallow and deep networks. Deep networks exploit multilevel symmetries through reflections that capture increasingly more complex relations between data as networks deepen. Every layer in a network creates more layers of reflective symmetry (see Figure 2).

## 1.1 RELATED WORK

Convexifying neural networks has been studied in (Bach, 2017; Bengio et al., 2005; Fang et al., 2019). However, these works assume that the network has infinite width, limiting their practical application. Recently, the training problem for finite-width, shallow ReLU networks has been shown to be equivalent to a convex reformulation with group norm regularization (Ergen & Pilanci, 2020; 2021; Pilanci & Ergen, 2020).

The use of geometric algebra for neuron operations is explored in (Brehmer et al., 2023; Ruhe et al., 2023). In contrast, our work uses conventional, real-valued neuron weights but interprets the features using geometric algebra. Pilanci (2023b) uses geometric algebra to formulate 2 and 3-layer ReLU neural network training problems as convex Lasso problems with features that represent high-dimensional volumes of polytopes spanned by training data. From the Lasso problem, an optimal network can be reconstructed, and its weights are orthogonal to the training data. The input data is of arbitrary dimension.

Zeger et al. (2024) shows that for 1-D input data and a variety of activation functions and arbitrarily deep networks, the training problem can be written as a Lasso problem. Moreover, for absolute value activations, simple networks with 1-D input exhibit features that represent reflections of training data, demonstrating that neural networks learn geometric features.

In this paper, we extend Pilanci (2023a) and Zeger et al. (2024) to prove that neural networks with absolute value activation and arbitrary input dimension and depth can be reformulated as equivalent Lasso problems with reflection features, using geometric algebra. The Lasso model suggests that larger models learn features related to symmetries in the data. This work differs from Pilanci (2023a) by considering absolute value instead of ReLU activation, which introduces new reflection features, and differs from Zeger et al. (2024) by considering arbitrary d-dimensional input instead of 1-D data. This work provides foundational analysis of networks with arbitrary dimensional data as functions that represent higher levels of reflections as networks deepen.

Unlike prior work, this work also reveals a *sparsity factor*, which is the ratio of $l_2$ and $l_1$ norms of a vector, in a network's equivalent Lasso model, which encourages parsimonious solutions. This sparsity factor has also been studied in (Yin et al., 2014; Xu et al., 2021). Additionally, this paper proves that the reflection complexity grows linearly with the number of layers in a deep narrow network. The reflection features suggest that networks such as Large Language Models (LLMs) may learn analogous structures and concepts in language. The presence of geometric structures in LLMs has been studied in (Park et al., 2024). Neural networks trained using LLM embeddings appear to learn the features predicted in Lasso models (Section 5).

## 1.2 CONTRIBUTIONS

We show the following:

- Training deep narrow networks of arbitrary depth is equivalent to solving convex Lasso problems with a finite set of features representing multilevel symmetries via reflections of increasing complexity (Theorem 4.4, Theorem 4.5).

- There are explicit Lasso dictionaries for 3-layer networks, and reconstructions of optimal networks from corresponding Lasso models. The reconstructed, optimal first-layer weight is orthogonal to a subset of augmented training data (Theorem 4.1).

- Networks with 3 layers can be described in terms of geometric algebra. They measure volumes spanned by training data and distances to reflection planes (Theorem 4.1).

- Neural networks trained using LLM text vector embeddings learn similar features as Lasso models (Section 5, Appendix C).

## 1.3 Notation

Let $e_i \in \mathbb{R}^d$ denote the $i^{\text{th}}$ canonical basis vector. Let $[d] = \{1, \cdots, d\}$. The number of non-zero elements in a vector $\mathbf{z}$ is $\|\mathbf{z}\|_0$. Neural network inputs, parameters, and outputs are all real-valued. $\text{Vol}(\mathbf{v}_1, \cdots, \mathbf{v}_d)$ is the unsigned volume of the polytope spanned by $\mathbf{v}_1, \cdots, \mathbf{v}_d$.

## 2 Background: Geometric Algebra

Consider the Euclidean vector space $\mathbb{R}^d$ with a scalar product (also called a inner or dot product for vector inputs). A *geometric algebra over* $\mathbb{R}^d$ is a vector space $\mathbb{G}^d$ (over the scalar field $\mathbb{R}$) of elements called *multivectors*, equipped with an associative binary operation called a *geometric product* that satisfies the following: (i) $\mathbb{G}^d$ contains $\mathbb{R}^d$ and is closed under the geometric product, (ii) geometric product is bilinear and distributes over addition, (iii) scalar multiplication is associative and commutative with the geometric product, (iv) the geometric product of a vector with itself is equal to the scalar product of the vector with itself Perwass et al. (2009).

The geometric product of $A, B \in \mathbb{G}^d$ is denoted as $AB$. It can be shown that the geometric product of the basis vectors $e_i, e_j$ of $\mathbb{R}^d \subset \mathbb{G}^d$ satisfy $e_i e_i = 1$ and $e_i e_j = -e_j e_i$ Perwass et al. (2009). Every multivector is a linear combination of *basis blades*, which are multivectors that are geometric products of the form $E = e_{\mathcal{G}[1]} \cdots e_{\mathcal{G}[|\mathcal{G}|]} = \prod_{i \in \mathcal{G}} e_i$, where $\mathcal{G} \subset [d]$ is an ordered set with no repeated elements and $\mathcal{G}[i]$ is its $i^{\text{th}}$ element. A basis blade $E$ has *grade* $g(E) = |\mathcal{G}|$. Scalars are defined to have grade 0. A dot product $A \cdot B$ and wedge product $A \wedge B$ are defined for multivectors in a way that generalizes the vector inner product and cross product, respectively (Appendix B). For vectors $A, B \in \mathbb{R}^d \subset \mathbb{G}^d$, $AB = A \cdot B + A \wedge B$ where the dot product represents the projection of $A$ onto $B$, and the wedge product represents the signed (oriented) area of the triangle spanned by $A$ and $B$.

The *generalized cross product* Berger (2009), Pilanci (2023a) of $\mathbf{v}_1, \cdots, \mathbf{v}_{d-1} \in \mathbb{R}^d$ is defined as

$$\times_{i=1}^{d-1} \mathbf{v}_i = \sum_i^{d-1} (-1)^{i-1} |V_i| e_i \tag{5}$$

where $V_i = [\mathbf{v}_1, \cdots, \mathbf{v}_{i-1}, \mathbf{v}_{i+1}, \cdots, \mathbf{v}_{d-1}]$ is a $(d-1) \times (d-1)$ square matrix and $e_i$ is the $i^{\text{th}}$ canonical basis vector. It can be shown that the generalized cross product is a vector that is orthogonal to $\mathbf{v}_1, \cdots, \mathbf{v}_{d-1}$. Since $e_i e_i = 1$ and $e_i e_j = -e_j e_i$, the grade of all multivectors in $\mathbb{G}^d$ is at most $d$. A *pseudoscalar* is a multivector of grade $d$. The *unit pseudoscalar* is $\mathbf{I} = e_1 \cdots, e_d$. The *unit pseudoscalar inverse* is $\mathbf{I}^{-1} = e_d \cdots e_1$ which satisfies $\mathbf{I}^{-1}\mathbf{I} = 1 = \mathbf{I}\mathbf{I}^{-1}$. The *Hodge star operator* $\star : \mathbb{G}^d \to \mathbb{G}^d$ is defined as $\star(\mathbf{v}) = \star \mathbf{v} = \mathbf{v}\mathbf{I}^{-1}$. If $\mathbf{v}$ is a wedge product of vectors, $\star \mathbf{v}$ represents the orthogonal complement of their span. It can be shown that the generalized cross product (5) can be equivalently written as

$$\times_{i=1}^{d-1} \mathbf{v}_i = \star \wedge_{i=1}^{d-1} \mathbf{v}_i. \tag{6}$$

It can be shown that the $l_2$ norm of the generalized cross product of $\mathbf{v}_1, \cdots, \mathbf{v}_{d-1}$ gives the $d-1$ volume of the polytope spanned by $\mathbf{v}_1, \cdots, \mathbf{v}_{d-1}$, while the inner product of $\times_{i=1}^{d-1} \mathbf{v}_i$ with a vector $\mathbf{v}_d$ gives the volume of the polytope spanned by $\mathbf{v}_1, \cdots, \mathbf{v}_d$ (Berger, 2009).

## 3 Prior work on shallow networks

This section introduces the equivalences established in prior work between simple 2-layer networks and Lasso problems. The Lasso problems have features representing volumes spanned by training data. The convex Lasso problem is *equivalent* to the non-convex training problem: they have the same optimal value and a network that is optimal in the training problem can be reconstructed from an optimal Lasso solution. (Zeger et al., 2024) shows that

**Theorem 3.1** ((Zeger et al., 2024))**.** *The training problem for a* 2*-layer deep narrow network and* 1*-D data is equivalent to a Lasso problem with dictionary elements*

$$\mathbf{A}_{i,j} = |x_i - x_j| \tag{7}$$

*provided that* $m \geq \|\mathbf{z}^*\|_0$, *where* $\mathbf{z}^*$ *is a solution to the Lasso problem.*

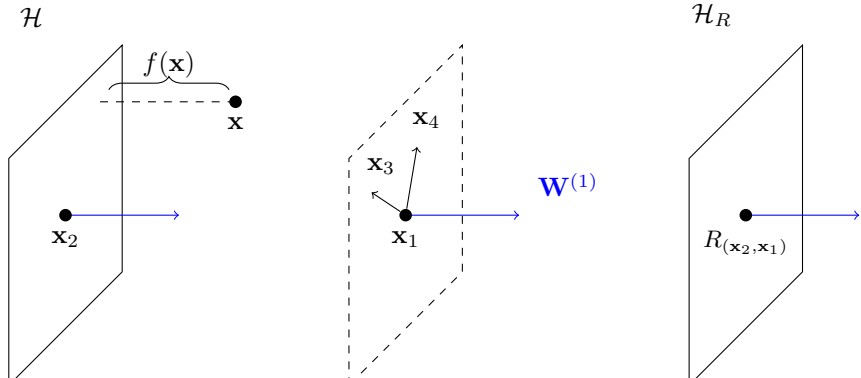

Figure 1: Example of reflection planes for 3-layer neural networks with data in $\mathbb{R}^3$.

The dictionary elements $|x_i - x_j|$ in Theorem 3.1 measure the distance between training samples $x_i$ and $x_j$, or equivalently, their 1-D volume. (Pilanci, 2023b) extends this volume interpretation to higher dimensions as

**Theorem 3.2** ((Pilanci, 2023b)). *The training problem for a* 2-*layer ReLU network with bias parameters set to* 0 *and data of arbitrary dimension is equivalent to a Lasso problem with dictionary elements*

$$\mathbf{A}_{i,j} = \frac{\mathrm{Vol}_+(\mathbf{x}_i, \mathbf{x}'_{j_1}, \cdots, \mathbf{x}'_{j_{d-1}})}{\left\| \mathbf{x}'_{j_1} \wedge \cdots \wedge \mathbf{x}'_{j_{d-1}} \right\|_1} = \frac{\left( \star \left( \mathbf{x}_i \wedge \mathbf{x}'_{j_1} \wedge \cdots \wedge \mathbf{x}'_{j_{d-1}} \right) \right)_+}{\left\| \mathbf{x}'_{j_1} \wedge \cdots \wedge \mathbf{x}'_{j_{d-1}} \right\|_1} \tag{8}$$

*where the multi-index* $j=(j_1,\cdots,j_{d-1})$ *indexes over all combinations of* $d-1$ *samples of* $\mathbf{x}_1,\cdots,\mathbf{x}_N, e_1,\cdots,e_d$.

In Theorem 3.2, $\mathrm{Vol}_+$ refers to the positive part of the signed volume. Similar results holds for other 2-layer networks, including those with nonzero bias, and with absolute value activation. This work extends the above results to deeper layers by analyzing deep narrow networks.

## 4 MAIN RESULTS

In this section, we state our main results on the equivalence between training deep neural networks and Lasso problems with geometric features. In contrast to shallow networks (Section 3), the deeper layers enable features to represent reflections of training data.

### 4.1 3-LAYER NEURAL NETWORKS

We show that 3-layer deep narrow networks are equivalent to Lasso problems with discrete, explicit dictionaries. Proofs in this section are deferred to Appendix E. The next result gives the equivalence between the training problem (2) for 3-layer deep narrow networks (3) and Lasso models (4), using the language of geometric algebra (Section 2). Let $\mathrm{Vol}(\mathbf{v}_1,\cdots,\mathbf{v}_d)$ denote the volume of the parallelotope spanned by $\mathbf{v}_1,\cdots,\mathbf{v}_d$.

**Theorem 4.1.** *The training problem for a* 3-*layer deep narrow network and* $d$-*dimensional data is equivalent to a Lasso problem with dictionary elements* $\mathbf{A}_{i,j} = f_j(\mathbf{x}_i)$ *defined as*

$$f_j(\mathbf{x}) = \frac{\left| \mathrm{Vol}\left( \mathbf{x} - \mathbf{x}'_{j_1}, \mathbf{x}_{j_2} - \mathbf{x}'_{j_3}, \cdots, \mathbf{x}_{j_{2(d-1)}} - \mathbf{x}'_{j_{2d-1}} \right) - \mathrm{Vol}\left( \mathbf{x}_{j_0} - \mathbf{x}'_{j_1}, \mathbf{x}_{j_2} - \mathbf{x}'_{j_3}, \cdots, \mathbf{x}_{j_{2(d-1)}} - \mathbf{x}'_{j_{2d-1}} \right) \right|}{\left\| \left( \mathbf{x}_{j_2} - \mathbf{x}'_{j_3} \right) \times \cdots \times \left( \mathbf{x}_{j_{2(d-1)}} - \mathbf{x}'_{j_{2d-1}} \right) \right\|_1}$$

$$\tag{9}$$

*provided that* $m \geq \|\mathbf{z}^*\|_0$, *where* $\mathbf{z}^*, \xi^*$ *is a Lasso solution. The multi-index* $j=(j_{-1}, j_0, j_1, j_2, \cdots, j_{2d-1})$ *indexes over all* $j_{-1}, j_{2k} \in [N], \mathbf{x}'_{j_1} \in \left\{ \frac{\mathbf{x}_{j_{-1}} + \mathbf{x}_{j_0}}{2}, \mathbf{x}_{j_{-1}} \right\}$ *and* $\mathbf{x}'_{j_{2k+1}} \in \left\{ \mathbf{x}'_{j_1}, R_{\left( \mathbf{x}_{j_0}, \mathbf{x}'_{j_1} \right)} \right\} \cup \{ \mathbf{x}_{j_{2k}} - e_l \}_{l \in [d]}$ *for* $k \geq 1$. *An optimal neural network can be recon-*

structed as $f_3(\mathbf{x}; \theta) = \sum_i z_i^* f_j(\mathbf{x}) + \xi^*$, and the corresponding optimal $\mathbf{W}^{(1)}$ is orthogonal to $\mathbf{x}_{j_2} - \mathbf{x}'_{j_3}, \cdots, \mathbf{x}_{j_{2(d-1)}} - \mathbf{x}'_{j_{2d-1}}$.

Theorem 4.1 gives a novel characterization of optimal neural networks as Lasso solutions. Theorem 4.1 states that we can train a globally optimal network by constructing a dictionary matrix, solving the Lasso problem (4) using well-known techniques (Efron et al., 2004), and then using a Lasso solution to reconstruct an optimal network. The dictionary is constructed from differences of points in an *augmented* training data set composed of reflections and averages of training data. For shorthand, we may refer to these augmented data points as the training data. Taking averages of training data has been used in effective data augmentation to improve the performance of training (Zhang et al., 2018). This Lasso equivalence may suggest theoretical reasons underlying this phenomenon. We now discuss several advantages of using the Lasso problem to train networks and address various aspects of the model equivalence.

**Training benefits:** The convexity of the Lasso problem ensures global optimality. In non-convex problems, training can get stuck in local, suboptimal optima depending on the initialization. The convex Lasso model avoids this problem. Additionally, using the Lasso model reduces the need to tune hyperparameters such as the number of neurons $m$, as $m = \|\mathbf{z}^*\|_0$ suffices.

**Uniqueness:** The training problem and Lasso problems may not have unique solutions. (Zeger et al., 2024) discusses the relationship between the solution set of the training problem with 1-D data and the set of neural networks reconstructed from all Lasso solutions. Analysis of the Lasso solution sets and stationary points in the training problem for higher dimensional data is an area for future work.

**Complexity:** The complexity of finding a Lasso solution with a dictionary of size $N \times F$ is $O(N^2 F)$ (Efron et al., 2004), which is linear in the number of feature vectors $F$. $F$ is finite and can be bounded.

**Lemma 4.2.** *The 3-layer Lasso dictionary $\mathbf{A}$ in Theorem 4.1 consists of $O\left((Nd)^d\right)$ feature vectors. The exponential complexity in $d$ can not be avoided for global optimization of ReLU networks unless* $\mathbf{P} = \mathbf{NP}$.

Lemma 4.2 is an overestimate of the dictionary size, since the Lasso dictionary contains repeated columns and the presence of linearly dependent vectors $\mathbf{x}_{j_k} - \mathbf{x}'_{j_{k+1}}$ (9) spanning a polytope will make its volume 0, resulting in a $\mathbf{0}$ vector that does not contribute to the Lasso model. To reduce the computational overhead of creating the dictionary, the dictionary features can be subsampled (Wang et al., 2024). (Wang et al., 2021) suggests that subsampling hyperplanes corresponds to arriving at a local optima. Another practical usage of the Lasso model is to use it in a polishing step instead of using it to train the entire model from scratch. Polishing consists of partially training a network with the non-convex training problem, extracting the breakplanes from the neurons to estimate a Lasso dictionary, and then using the Lasso estimate to fine-tune the network (Pilanci, 2023b).

**Comparison to ReLU**: (Zeger et al., 2024) shows that 2-layer networks with absolute value activation are equivalent to those with ReLU when the network uses a skip connection, and increasing the width of ReLU networks with 1-D input data introduces reflection features. Figure 12 illustrates a reflection feature occurring in a network trained on 2-D data. This suggests that wider ReLU architectures with higher-dimensional data will have reflection features, and is an area for future analysis.

**Interpretibility:** The $l_1$ regularization in the Lasso problem selects a minimal number of feature vectors which are discrete samples of *features* $f_j(\mathbf{x})$. Intuitively, the network *learns* these features, as a linear combination of feature functions corresponds to a parsimonious and optimal network. The features interpolate subsets of the training data. The mapping between the features and optimal weights is in the Appendix (Definition E.2). Theorem 4.1 gives a volumetric interpretation of features. The features also have geometric algebraic and a distance-based interpretations.

A vector $\mathbf{x}$ is *sparse* if it has few non-zero elements. The *sparsity factor* of $\mathbf{x} \neq \mathbf{0}$ is $r(\mathbf{x}) = \frac{\|\mathbf{x}\|_2}{\|\mathbf{x}\|_1} \in \left[\frac{1}{\sqrt{N}}, 1\right]$. The sparsity factor is a measure of a vector's sparseness. If $\mathbf{x}$ has one non-zero element, then $r(\mathbf{x}) = 1$ and if $\mathbf{x}$ is parallel to $\mathbf{1}$, then $r(\mathbf{x}) = \frac{1}{\sqrt{N}}$. The more sparse $\mathbf{x}$ is, the larger its sparsity factor. Next, given $j_{-1}, j_0 \in [N], \mathbf{w} \in \mathbb{R}^d$, define the hyperplanes $\mathcal{H} = \{\mathbf{x} \in \mathbb{R}^d : (\mathbf{x} - \mathbf{x}_{j_{-1}})\mathbf{w} = 0\}, \mathcal{H}_0 = \{\mathbf{x} \in \mathbb{R}^d : (\mathbf{x} - \mathbf{x}_{j_0})\mathbf{w} = 0\}$. The *average* of the hyperplanes $\mathcal{H}$ and $\mathcal{H}_0$ is the hyperplane $\mathcal{H}_A = \left\{\mathbf{x} \in \mathbb{R}^d : \left(\mathbf{x} - \frac{\mathbf{x}_{j_{-1}} + \mathbf{x}_{j_0}}{2}\right)\mathbf{w} = 0\right\}$. The *reflection* of the hyperplanes $\mathcal{H}$

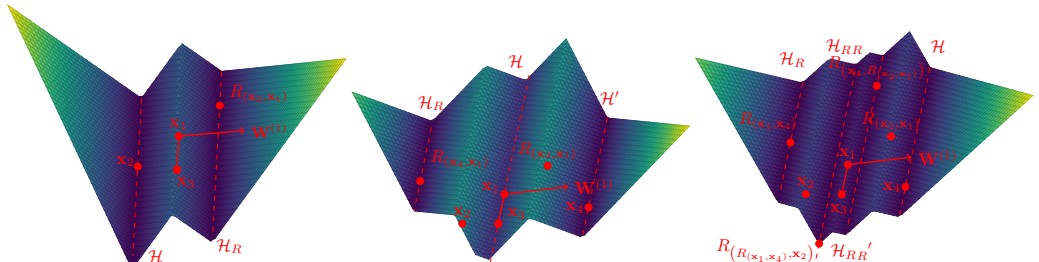

Figure 2: Examples of 3-layer (left) and 4-layer (middle, right) deep narrow network features for 2-D data. Red dashed lines indicate reflection planes (lines in $\mathbb{R}^2$).

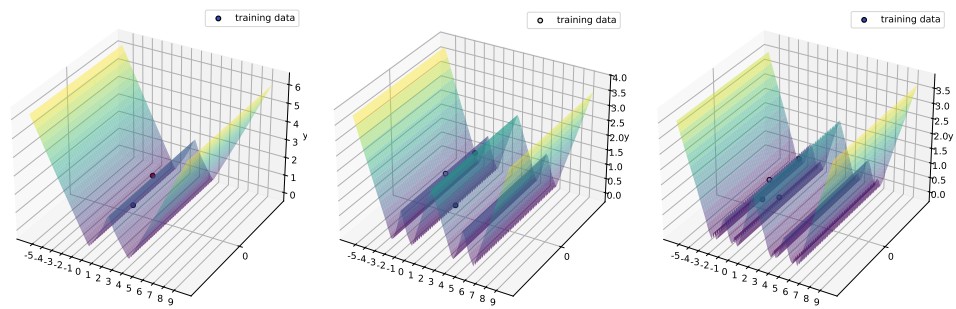

Figure 3: Adam-trained neural net with absolute value activation and 2-D training data, of depth 3 (left), 4 (middle), and 5 (right). There are breakplanes at reflections of up to order $L-2$ of data points. The 3 and 4-layer networks match the corresponding features shown in the left and right-most plots of Figure 2, respectively.

across $\mathcal{H}_0$ is $\mathcal{H}_R = \left\{ \mathbf{x} \in \mathbb{R}^d : \left( \mathbf{x} - R_{\left( \mathcal{H}_{j-1}, \mathcal{H}_{j_0} \right)} \right) \mathbf{w} = 0 \right\}$. These definitions will be used in the next result to describe network features.

**Theorem 4.3.** *The* 3*-layer deep narrow network features in Theorem 4.1 can be written as*

$$f_j(\mathbf{x}) = \frac{\left\| \left| \star \left( \mathbf{x} - \mathbf{x}'_{j_1} \wedge \mathbf{x}_{j_2} - \mathbf{x}'_{j_3} \wedge \cdots \wedge \mathbf{x}_{j_{2(d-1)}} - \mathbf{x}'_{j_{2d-1}} \right) \right| - \left| \star \left( \mathbf{x}_{j_0} - \mathbf{x}'_{j_1} \wedge \mathbf{x}_{j_2} - \mathbf{x}'_{j_3} \wedge \cdots \wedge \mathbf{x}_{j_{2(d-1)}} - \mathbf{x}'_{j_{2d-1}} \right) \right| \right\|}{\left\| \left( \mathbf{x}_{j_2} - \mathbf{x}'_{j_3} \right) \wedge \cdots \wedge \left( \mathbf{x}_{j_{2(d-1)}} - \mathbf{x}'_{j_{2d-1}} \right) \right\|_1}$$

$$= r(\mathbf{w}) \begin{cases} \mathrm{Dist}\left( \mathbf{x}, \mathcal{H} \cup \mathcal{H}_R \right) & \text{if } \mathbf{x}'_{j_1} = \mathbf{x}_{j-1} \\ \mathrm{Dist}\left( \mathbf{x}, \mathcal{H} \cup \mathcal{H}_A \right) & \text{else} \end{cases}$$

(10)

*where* $\mathbf{w} = \left( \left( \mathbf{x}_{j_2} - \mathbf{x}'_{j_3} \right) \times \cdots \times \left( \mathbf{x}'_{j_{2(d-1)}} - \mathbf{x}'_{j_{2d-1}} \right) \right)^T$. *In the reconstructed optimal network,* $\mathbf{W}^{(1)}$ *is a scalar multiple of* $\mathbf{w}$.

Theorem 4.3 shows that neural network features represent distances to parallel planes containing averages and reflections of training data. Moreover, these planes are orthogonal to $\mathbf{W}^{(1)}$ and are spanned by a subset of training data. Theorem 4.3 also describes the features as wedge products, and gives a formula to explicitly compute the dictionary elements using wedge products or generalized cross products (5), (6).

Theorem 4.3 shows that the Lasso dictionary includes *reflection features* that measure the minimum distance to two parallel *reflection planes*: $\mathcal{H}$, which contains $\mathbf{x}_{j-1}$, and $\mathcal{H}_R$, which is the reflection of $\mathcal{H}$ across the hyperplane $\mathcal{H}_0$ containing $\mathbf{x}_{j_0}$. Since distance is an unsigned value, the network induces a symmetry about the two planes. Figure 1 illustrates an example of reflection planes $\mathcal{H}, \mathcal{H}_R$ in $\mathbb{R}^3$. The right plot of Figure 2 graphs an example feature $f_j(\mathbf{x})$ in $\mathbb{R}^2$. The feature outputs the distance between the reflection "planes" $\mathcal{H}, \mathcal{H}_R$, which are lines in $\mathbb{R}^2$ and are plotted as red, dashed lines. These are axes of symmetry encoded in 3-layer features. In the notation used in Theorem 4.1,

Figure 1 and the right plot of Figure 2 both graph the case where $\mathbf{x}'_{j_1}=\mathbf{x}_1, \mathbf{x}_{j_0}=\mathbf{x}_2, \mathbf{x}_{j_2}=\mathbf{x}_3, \mathbf{x}'_{j_3}=\mathbf{x}_1$, (and additionally $\mathbf{x}_{j_4}=\mathbf{x}_4, \mathbf{x}'_5=\mathbf{x}_1$ for Figure 1).

We can interpret reflection planes as *concepts*. The concepts collect training data into planes, and the neural network measures the distance, or similarity, between inputs and concepts. The concepts are based at a training data point, and a point that is not necessarily in the training set: a reflection of a data point about another data point. In other words, the neural network predicts single-level reflective symmetries in the data. Intuitively, these symmetries can act as a translation between analogous concepts across different domains, with reflections representing shifts in structure while preserving core functionality.

Another way to view the geometry of a feature is by analysing its breakplanes. A piecewise linear function $f : \mathbb{R} \to \mathbb{R}$ has a *breakpoint* at $\mathbf{x}$ if $f$ changes slope at $\mathbf{x}$. A function $g : \mathbb{R}^{1\times d} \to \mathbb{R}$ defined by $g(\mathbf{x})=f(\mathbf{xw})$ where $\mathbf{w}\in\mathbb{R}^d$ has a *breakplane* along the plane $\{\mathbf{x} \in \mathbb{R}^{1\times d} : \mathbf{xw}+b=0\}$ if $f$ has a breakpoint at $\mathbf{xw}$. A breakplane of a function is a "kink" in its graph. As seen above, the reflection hyperplanes are breakplanes of 3-layer networks and create axes of symmetry. The reflection breakplanes in 3-layer networks generalize to deeper networks, as shown next.

## 4.2 DEEPER NETWORKS

Here, we extend a Lasso equivalence to deeper networks. For $L \geq 2$, the $L$-layer *feature function* is a parameterized function $f_j : \mathbb{R}^d \to \mathbb{R}$ defined as

$$f_j(\mathbf{x}) = \left| \cdots \left| \left| \left| \mathbf{x}\mathbf{W}^{(1)} + b^{(1)} \right| + b^{(2)} \right| \cdots \right| + b^{(L-1)} \right|. \tag{11}$$

A feature function (11) can be viewed as a basic unit of a deep narrow network (3), or a deep narrow network where $m=1, \alpha=1, \xi=0$ and $w^{(i,l)}=1$ for $l>1$. A feature function has *data feature biases* if for $n^{(1)}=\mathbf{x}'_{j_1}, n^{(2)}, \cdots, n^{(L-1)}\in[N]$, all of its bias parameters are defined recursively as

$$b^{(1)}= -\mathbf{x}'_{j_1}\mathbf{W}^{(1)}, \quad b^{(l)}= -\left| \cdots \left| \left| \left| \mathbf{x}_{n^{(l)}}\mathbf{W}^{(1)} + b^{(1)} \right| + b^{(2)} \right| \cdots \right| + b^{(l-1)} \right|, \tag{12}$$

for $l\in[L-1]$. The *data feature sub-library* is a set of feature functions with data feature biases. The next result states that deep narrow networks of *any* depth are equivalent to Lasso problems.

**Theorem 4.4.** *The training problems for deep narrow networks with arbitrary depth and input dimension are equivalent to Lasso problems with finite, discrete dictionaries. The set of feature vectors contains a data feature sub-library and consists of feature functions* (11) *sampled at training data points. An optimal network is reconstructed as* $\sum_i z_i^* f_j(\mathbf{x}) + \xi^*$.

The proof is in Appendix E. Definition E.2 describes a mapping between features and optimal weights. Explicit expressions of features written as (11) for 2 and 3-layer networks is in Remark E.1. The data feature sub-library is a set of novel features representing reflections of increasing complexity, or *order*. The order of a reflection is defined recursively as follows. An *order*-0 reflection of a point $\mathbf{x}_0\in\mathbb{R}^d$ is simply $R_0(\mathbf{x}_0)=\mathbf{x}_0$. Given points $\mathbf{x}_0, \mathbf{x}_1$, the *standard reflections* $R_{(\mathbf{x}_0,\mathbf{x}_1)}$ and $R_{(\mathbf{x}_1,\mathbf{x}_0)}$ are *order*-1 reflections, which create a single-level symmetry of the point being reflected and its reflection. An *order*-2 reflection is a reflection of a reflection, which induces a *second-level symmetry* around the reflection. In general for $k>0$, the *order*-$k$ *reflection* of $\mathbf{x}_0, \cdots, \mathbf{x}_k\in\mathbb{R}^d$ is of the form $R_k(\mathbf{x}_0, \cdots, \mathbf{x}_k)\in\left\{R_{(R_{k-1}(\mathbf{x}_0,\cdots,\mathbf{x}_{k-1}),\mathbf{x}_k)}, R_{(\mathbf{x}_k,R_{k-1}(\mathbf{x}_0,\cdots,\mathbf{x}_{k-1}))}\right\}$. Figure 4 plots examples of reflections of order $0, 1$ and $2$ in $\mathbb{R}^2$. As seen in Figure 4, in $\mathbb{R}^2$, $R_{(\mathbf{a},\mathbf{b})}$ is the point on the line between $\mathbf{a}$ and $\mathbf{b}$, whose distance from $\mathbf{b}$ is the same as the distance from $\mathbf{b}$ to $\mathbf{a}$, creating a symmetry.

Theorem 4.3 states that the features for a 3-layer network measure distances to reflection planes, which include breakplanes at first-order reflections. The left plot of Figure 2 illustrates an example of a 3-layer feature. The weight $\mathbf{W}^{(1)}$ is orthogonal to $\mathbf{x}_3-\mathbf{x}_1$ and the breakplanes occur along $\mathcal{H}_1=\left\{(\mathbf{x}-\mathbf{x}_1)\mathbf{W}^{(1)}=0\right\}, \mathcal{H}=\left\{(\mathbf{x}-\mathbf{x}_2)\mathbf{W}^{(1)}=0\right\}$ and $\mathcal{H}_R = \left\{\left(\mathbf{x}-R_{(x_2,\mathbf{x}_1)}\right)\mathbf{W}^{(1)}=0\right\}$.

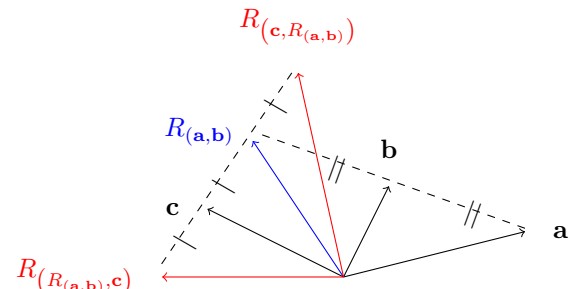

Figure 4: Reflections in $\mathbb{R}^2$ of different orders. Order 0: $\mathbf{a}, \mathbf{b}, \mathbf{c}$. Order 1: $R_{(\mathbf{a},\mathbf{b})}$. Order 2: $R_{\left(R_{(\mathbf{a},\mathbf{b})},\mathbf{c}\right)}, R_{\left(\mathbf{c},R_{(\mathbf{a},\mathbf{b})}\right)}$ .

The height of the graph is the minimum of the distance from the reflection planes $\mathcal{H}$ and $\mathcal{H}_R$. Similarly, 4-layer features can have breakplanes that represent first or second-order reflections, as shown respectively in the middle and right plots of Figure 2, which plots examples of data feature sub-library features. The peaks and troughs of the features plotted in Figure 2 are breakplanes. The troughs are indicated by dashed red lines and depict higher-level symmetries and reflection planes. Deeper networks have the capacity for learning increasingly complex reflection features and multilevel symmetries.

**Theorem 4.5** (Multilevel symmetries). *A deep narrow network of arbitrary input dimension and depth $L$ is equivalent to a Lasso problem with a dictionary containing features with no reflections for $L = 2$, standard reflections for $L=3$, and up to order-$2(L-3)$ reflections for $L > 3$.*

The proof is in Appendix F. Theorem 4.5 shows that each additional layer in a network induces higher order reflections and deeper levels of symmetry. Theorem 4.5 describes the data feature sub-library in Theorem 4.4 and shows that the maximum order of reflections in features grows linearly with depth. A full analysis of the entire dictionary for networks with more than 3 layers is an area for future work. Nonetheless, our approaches and proof techniques demonstrate the principle of how to find all of the dictionary elements for deeper networks, as discussed in the proof of Theorem 4.4.

## 5 NUMERICAL RESULTS

We perform experiments training the *standard* network

$$f_L\left(\mathbf{x};\theta\right)=\sigma\left(\cdots\left(\sigma\left(\sigma\left(\mathbf{x}\mathbf{W}^{(1)}+\mathbf{b}^{(1)}\right)\mathbf{W}^{(2)}+\mathbf{b}^{(2)}\right)\cdots\right)\mathbf{W}^{(L-1)}+\mathbf{b}^{(L-1)}\right)\boldsymbol{\alpha}+\xi. \quad (13)$$

where $\mathbf{W}^{(1)}\in\mathbb{R}^{d\times 1}, \mathbf{b}^{(1)}\in\mathbb{R}, \cdots, \mathbf{W}^{(l)}\in\mathbb{R}, \mathbf{b}^{(l)}\in\mathbb{R}, \mathbf{W}^{(L-1)}\in\mathbb{R}^{1\times m}, \mathbf{b}^{(L-1)}\in\mathbb{R}^{1\times m}, \xi\in\mathbb{R}, \boldsymbol{\alpha}\in\mathbb{R}^m$. For 2-layer networks, the standard network (13) is equivalent to the network (1), and (1) can be converted into a standard architecture (Zeger et al., 2024). The standard architecture is more traditional, and we perform experiments on the standard architecture to demonstrate that the Lasso model can be useful for this architecture as well.

### 5.1 SIMULATED DATA

In Figure 3, 3, 4, and 5-layer networks (13) are trained with Adam. The second coordinate is 0 in all samples. The data is given in Appendix C. We first project the 2-D data to 1-D along the first coordinate and solve the Lasso problem for the 1-D data as $\beta\rightarrow 0$. The *minimum ($l_1$) norm subject to interpolation* version of the Lasso problem is

$$\min_{\mathbf{z},\xi} \|\mathbf{z}\|_1 \text{ s.t. } \mathbf{A}\mathbf{z} + \xi\mathbf{1} = \mathbf{y}. \quad (14)$$

Loosely speaking, as $\beta\rightarrow 0$, if $\mathbf{A}$ has full column rank, the Lasso problem approaches the minimum norm problem (14). For 1-D data, an optimal solution to (14) for certain simple sets of training data is known (Zeger et al., 2024). After solving the Lasso problem, the non-convex model (2) with the

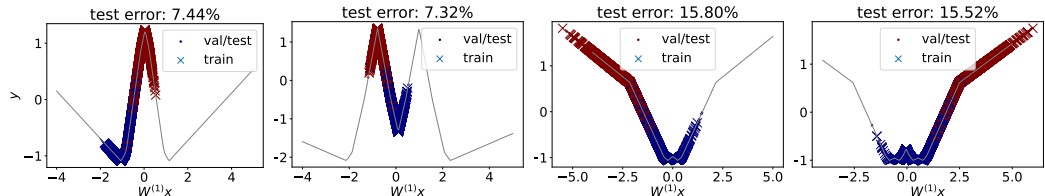

Figure 5: 3-layer, followed by 4-layer (right) deep narrow networks trained on text embeddings of IMDB reviews outputted by OpenAI GPT4 (left-most 2 plots) and Bert (right-most 2 plots). The network's predictions are plotted as a function of the input $\mathbf{x}$ projected to 1-D as $\mathbf{W}^{(1)}\mathbf{x}$. True labels of 1 and $-1$ are indicated by red and blue, respectively. The networks exhibit multi-level symmetry.

orignal 2-D data is trained with $\beta = 10^{-7} \approx 0$, where a minimum norm problem solution is used to pre-initialize a subset of the neurons. All other weights (we use $m=100$) are initialized randomly according to Pytorch defaults. Only one neuron per layer is pre-initialized, and for the neuron weight in the first layer, the second coordinate is left to random initialization. A learning rate of $5(10^{-3})$ and weight decay of $10^{-4}$ are used. The data is trained over $10^3$ epochs. A similar experiment is performed with a ReLU network, which also demonstrates breakplanes at reflections of training data (Appendix C Figure 12). As shown in Figure 3, the networks exhibit breakplanes at data points and their reflections which are not among the training data. Additionally, the neural networks exhibit features matching the shapes of those shown in the right and left plots of Figure 2 for 3 and 4 layers, respectively. The 5-layer network shows that features gain more complex reflections with depth.

## 5.2 LARGE LANGUAGE MODEL EMBEDDINGS

We also train neural networks using Adam to perform binary $(y_n = \pm 1)$ classification of text and observe multi-level symmetries. Models from OpenAI GPT4 and Bert are trained to embed text as vectors, which deep narrow networks are then trained on as input to classify the corresponding text. The deep narrow networks are trained using the non-convex regression problem (2). Positive network outputs are interpreted as $y_n = 1$ and negative outputs as $y_n = -1$.

Figure 5 plots predictions of neural nets that are trained to perform sentiment analysis, rating IMDB reviews as having positive or negative sentiments (Maas et al., 2011). Figure 5 plots the points $\left(\mathbf{W}^{(1)}\mathbf{x}_n, \hat{y}_n\right)$ in red and blue, where $\hat{y}_n = f_L(\mathbf{x}_n; \theta)$ is the network's prediction on the training sample $\mathbf{x}_n$ and the color corresponds to the training labels as red for $y_n = 1$ and blue for $y_n = -1$. The network prediction on all points $\mathbf{x}$ is plotted in gray. For any constant $c \in \mathbb{R}$, a Lasso feature (11) has constant value along $\{\mathbf{x} : \mathbf{W}^{(1)}\mathbf{x} = c\}$, so projecting the input along $\mathbf{W}^{(1)}$ can be viewed as a cross-section of the feature.

In general, there are (possibly sub-optimal) weights and biases that can make a deep narrow network be an asymmetrical function with many breakplanes that do not contain training samples. However, the trained networks in Figure 5 appear multilevel symmetric with breakplanes at data samples and resembles the shapes of the 3 and 4-layer reflection features illustrated in Figure 2. In particular, the 3-layer networks shown in the first and third plots from the left in Figure 5 resemble the 3-layer network in left plot of Figure 2, while the 4-layer network shown in the second and fourth plots in Figure 5 resemble the 4-layer network shown in the middle and right plots, respectively, of Figure 2. This is consistent with the sparse selection of features in the Lasso problem. Appendix C contains training details and additional results. Code from the github repository for Wang et al. (2024) was used to generate the embeddings.

## 6 CONCLUSION

We prove an equivalence between neural networks and Lasso problems with novel geometric features. Our convexification approach can be extended to other piecewise linear activation functions Zeger et al. (2024). A limitation of this work is the choice of the activation function and the $\ell_1$ regularization of the weights needed for analytic tractability. We believe that multi-level symmetries hold for standard ReLU networks, which is left for future work.

## 7 REPRODUCIBILITY STATEMENT

Proofs of results proved in prior work:

- Theorem 3.1: proof in (Zeger et al., 2024)

- Theorem 3.2: proof in (Pilanci, 2023b)

Proof of results proved in this work:

- Theorem 4.1: proof in Appendix E

- Lemma 4.2: proof in Appendix E

- Theorem 4.3: proof in Appendix E

- Theorem 4.4: proof in Appendix E

- Theorem 4.5: proof in Appendix F

Code is in the supplementary file. Please run "run.ipynb" to generate figures referenced in Section 5 and Appendix C.

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

# A   APPENDIX

# B   GEOMETRIC ALGEBRA

**Wedge and inner product**: A dot product of multivectors, unlike a dot product of vectors, can have multivector outputs. The *dot* or *inner product* of arbitrary basis blades $E_1 = \prod_{i \in \mathcal{G}^{(1)}} e_i$ and $E_2 = \prod_{i \in \mathcal{G}^{(2)}} e_i$ is $E_1 \cdot E_2 = E_1 E_2$ if $g(E_1 E_2) = |g(E_1) - g(E_2)|$, and 0 otherwise. In other words, if $G^{(1)} \subset G^{(2)}$, then $E_1 E_2$ is composed of basis blades in $\mathcal{G}^{(2)} - \mathcal{G}^{(1)}$, and $E_1 E_2 = 0$ otherwise. (Similarly if $G^{(2)} \subset G^{(1)}$). The *wedge* or *outer product* is defined by $E_1 \wedge E_2 = E_1 E_2$ if $E_1 E_2$ has grade $g(E_1 E_2) = g(E_1) + g(E_2)$, and 0 otherwise. So if $\mathcal{G}^{(i)} \cap \mathcal{G}^{(j)} = \emptyset$, then $E_1 E_2$ is composed of basis blades in $\mathcal{G}^{(1)} \cup \mathcal{G}^{(2)}$, and otherwise, $E_1 E_2 = 0$. The dot and wedge product extend to multivectors linearly. A *vector* is a special case of a multivector $\mathbf{v} \in \mathbb{R}^d \subset \mathbb{G}^d$ of grade 1. For more detailed background, see Perwass et al. (2009).

Note that linearly distributing terms and applying $e_i e_i = 1$ shows that $\wedge_{i=1}^{d-1} \mathbf{v}_i = \wedge_{i=1}^{d-1} \sum_{j=1}^{d} v_{ij} e_j$ consists of a sum of $d - 1$-vectors, so their product with $e_d \cdots e_1$, as implied by the Hodge star operator in (6) is a vector, which is consistent with (5).

**Remark B.1.** *It holds that* $\star \left( \mathbf{u} \wedge \left( \wedge_{i=1}^{d-1} \mathbf{v}_i \right) \right) = \mathbf{u}^T \left( \times_{i=1}^{d-1} \mathbf{v}_i \right) \in \mathbb{R}$ *and* $\| \wedge_{i=1}^{d-1} \mathbf{v}_i \|_2 = \left\| \times_{i=1}^{d-1} \mathbf{v}_i \right\|_2$. *This is because*

$$
\begin{aligned}
\wedge_{i=1}^{d-1} \mathbf{v}_i &= \sum_{k=1}^{d} \left( \left( \wedge_{i=1}^{d-1} \mathbf{v}_i \right)^T \left( e_1 \cdots e_{k-1} e_{k+1} \cdots e_d \right) \right) e_1 \cdots e_{k-1} e_{k+1} \cdots e_d \\
&= \sum_{k=1}^{d} \left( \times_{i=1}^{d-1} \mathbf{v}_i \right)_k e_1 \cdots e_{k-1} e_{k+1} \cdots e_d
\end{aligned}
\tag{15}
$$

*and* $e_j \wedge e_k = 0$ *if* $j \neq k$, *we have* $\mathbf{u} \wedge \left( \wedge_{i=1}^{d-1} \mathbf{v}_i \right) = \sum_{j=1}^{d} u_j e_j \wedge \left( \wedge_{i=1}^{d-1} \mathbf{v}_i \right) = \sum_{j=1}^{d} u_j e_j \wedge \left( \left( \times_{i=1}^{d-1} \mathbf{v}_i \right)_k \right) e_1 \cdots e_{j-1} e_{j+1} \cdots e_d$. *So* $\star \left( \mathbf{u} \wedge \left( \wedge_{i=1}^{d-1} \mathbf{v}_i \right) \right) = \mathbf{u} \wedge \left( \wedge_{i=1}^{d-1} \mathbf{v}_i \right) e_d \cdots e_1 = \mathbf{u}^T \left( \times_{i=1}^{d-1} \mathbf{v}_i \right) \in \mathbb{R}$. *From* (15) *it also follows that* $\| \wedge_{i=1}^{d-1} \mathbf{v}_i \|_2 = \left\| \times_{i=1}^{d-1} \mathbf{v}_i \right\|_2$.

**Remark B.2.** *It holds that*

$$
\left| \star \left( \mathbf{v}_1 \wedge \cdots \wedge \mathbf{v}_d \right) \right| = \mathrm{Vol}(\mathbf{v}_1, \cdots, \mathbf{v}_d)
\tag{16}
$$

*and*

$$
Dist \left( \mathbf{v}_d, Span(\mathbf{v}_1, \cdots, \mathbf{v}_d) \right) = \frac{\star (\mathbf{v}_1 \wedge \cdots, \wedge \mathbf{v}_d)}{\| \mathbf{v}_1 \wedge \cdots, \wedge \mathbf{v}_d \|_2}.
\tag{17}
$$

*Here,* $\mathbf{v}_1, \mathbf{v}_2$, *let* $Dist(\mathbf{v}_1, \mathbf{v}_2)$ *denote the unsigned Euclidean distance* $\| \mathbf{v}_1 - \mathbf{v}_2 \|_2$ *between* $\mathbf{v}_1$ *and* $\mathbf{v}_2$. *The distance between a vector* $\mathbf{v}$ *and a set* $S$ *is* $\min \{ Dist(\mathbf{v}, \mathbf{x}) : \mathbf{x} \in S \}$. *In other words,* $\mathbf{v}_1 \wedge \cdots \wedge \mathbf{v}_{d-1}$ *is a vector whose magnitude is the unsigned volume of the* $d-1$*-dimensional polytope spanned by* $\mathbf{v}_1, \cdots \mathbf{v}_{d-1}$, *which represents the "base" of the d-dimensional polytope* $\mathcal{P}(\mathbf{v}_1, \cdots, \mathbf{v}_d)$ *and* (17) *is the signed "height" of* $\mathcal{P}(\mathbf{v}_1, \cdots, \mathbf{v}_d)$, *or the distance of* $\mathbf{v}_d$ *to the span of* $\mathbf{v}_1, \cdots \mathbf{v}_d$. . *Note the Hodge star operator* $\star$ *converts pseudoscalars into a scalars.*

# C   NUMERICAL RESULTS

## C.1   SIMULATED DATA

In Figure 3, the training data is $(\mathbf{x}_1, y_1 = 2), (\mathbf{x}_2, y_2 = 0)$ for 3 layers, $(\mathbf{x}_1, y_1 = 2), (\mathbf{x}_2, y_2 = 0), (\mathbf{x}_3, y_3 = -1)$ for 4 layers, and $(\mathbf{x}_1, y_1 = 1.75), (\mathbf{x}_2, y_2 = 0.25), (\mathbf{x}_3, y_3 = 0.75), (\mathbf{x}_4, y_4 = -1.75)$ for 5 layers, where $\mathbf{x}_1 = (2, 0), \mathbf{x}_2 = (0, 0), \mathbf{x}_3 = (-1, 0), \mathbf{x}_4 = (-1.75, 0)$.

In Figure 12, the same experiment setup as used for Figure 3 is used for a 3-layer network, except $\mathbf{W}^{(1)} \in \mathbb{R}^{d \times 2}, \mathbf{b}^{(2)} \in \mathbb{R}, \mathbf{W}^{(2)} \in \mathbb{R}^{2 \times m}, \mathbf{b}^{(L-1)} \in \mathbb{R}^{2 \times m}$ and we require all neuron weights in each layer

have the same magnitude. The data is $\mathbf{x}_1 = (4,0), \mathbf{x}_2 = (3,0), \mathbf{x}_3 = (1,0), \mathbf{x}_4 = (0,0), \mathbf{x}_5 = (-1,0), y_1 = 2, y_2 = 1, y_3 = 0, y_4 = 0, y_5 = 1$. More general architectures are an area of exploration in future work.

## C.2 Large Language Model data

We perform the same experiments as in Section 5.2, but with the GLUE data set, specifically GLUE-CoLA and GLUE-QQP (Wang et al., 2019; Warstadt et al., 2018; Socher et al., 2013; Dolan & Brockett, 2005; Agirre et al., 2007; Williams et al., 2018; Rajpurkar et al.; Dagan et al., 2006; Bar Haim et al., 2006; Giampiccolo et al., 2007; Levesque et al., 2011; Bentivogli et al., 2009). GLUE (General Language Understanding Evaluation) is a benchmark for testing the performance of models learning language processing. GLUE-CoLA (Corpus of Linguistic Acceptability) contains text that is to be binary classified whether it is grammatically correct or not. GLUE-QQP (Quora Question Pairs) contains pairs of questions that are to be binary classified as having the same semantic meaning or not. Figure 6, Figure 7, Figure 8, Figure 9, Figure 10, Figure 11 plot the results. Vector embeddings from both GPT4 and Bert (Devlin et al., 2019) LLMs are used. As in Figure 5, points with $y_n=1$ are plotted in red, and points with $y_n=-1$ are plotted in blue. The overall network is plotted in gray. The training data is plotted with 'x' and the validation and test data are plotted with '.' markers.

A standard architecture (13) was trained using Adam with a learning rate of $5(10^{-3})$, weight decay of $10^{-4}$, and $\beta=10^{-7}$. There were $N=10^4$ data points, the network had $m=10$ neurons, and the network was trained over 100 epochs.

For OpenAI's GLUE-ColA input embeddings, 175 epochs and $m = 10$ were used. For Bert GLUE-ColA, 125 epochs and $m = 5$ were used. For OpenAI and Bert GLUE-QQP, 150 epochs and $m = 10$ were used. For OpenAI and Bert IMDB, 100 epochs and $m = 10$ were used. These parameters were chosen based on validation set results.

The networks are trained on the non-convex problem, and their performance is comparable to results in (Wang et al., 2024). What distinguishes this experiment is demonstrating that networks learn novel features consistent with Lasso models.

Networks trained on both OpenAI and Bert embeddings exhibit similarities to Lasso features Figure 3. However, the networks trained on OpenAI embeddings (Figure 6) have higher accuracy and closer matches to Lasso features than those trained using Bert. The higher-performing networks trained using GPT4, such as Figure 6, have striking similarities to Lasso features Figure 3. This is consistent with the network's equivalence to the Lasso model, as a network that is optimal in the Lasso problem is globally optimal. The figures show that when networks get deeper, they change by increasing their breakplanes, with more breakpoints appearing in the plots.

## D Neural net architecture

**Note:** The notation and some techniques for convexification are similar to (Zeger et al., 2024) and (Pilanci, 2023b). For finding the Lasso features, we can assume that the data matrix is full column rank (Pilanci, 2023b). All of the Lasso equivalences hold provided that $m^* \geq \|\mathbf{z}\|_0$.

Let $L \geq 2, m_0 = d, m_{L-1} = 1$ and $m_l \in \mathbb{N}$ for $l \in [L] - \{L-1\}$. A neural network (1) can be recursively defined as

$$f_L(\mathbf{x}; \theta) = \sum_{i=1}^{m_L} \mathbf{X}^{(i,L)} w^{(i,L)} + \xi,$$

where $\mathbf{X}^{(i,L)}$ is defined recursively as

$$\mathbf{X}^{(i,l+1)} = \sigma\left(\mathbf{X}^{(i,l)}\mathbf{W}^{(i,l)} + \mathbf{b}^{(i,l)}\right), \tag{18}$$

with initial condition

$$\mathbf{X}^{(i,1)} = \mathbf{x}.$$

We can view $\mathbf{X}^{(i,l)}$ as an input to to the $i^{\text{th}}$ *unit* in layer $l$. Note that for $i \in [m_L], l \in [L]$, we have $\mathbf{X}^{(i,l)} \in \mathbb{R}^{1 \times m_{l-1}}$. The set of regualarized parameters is $\theta_w =$

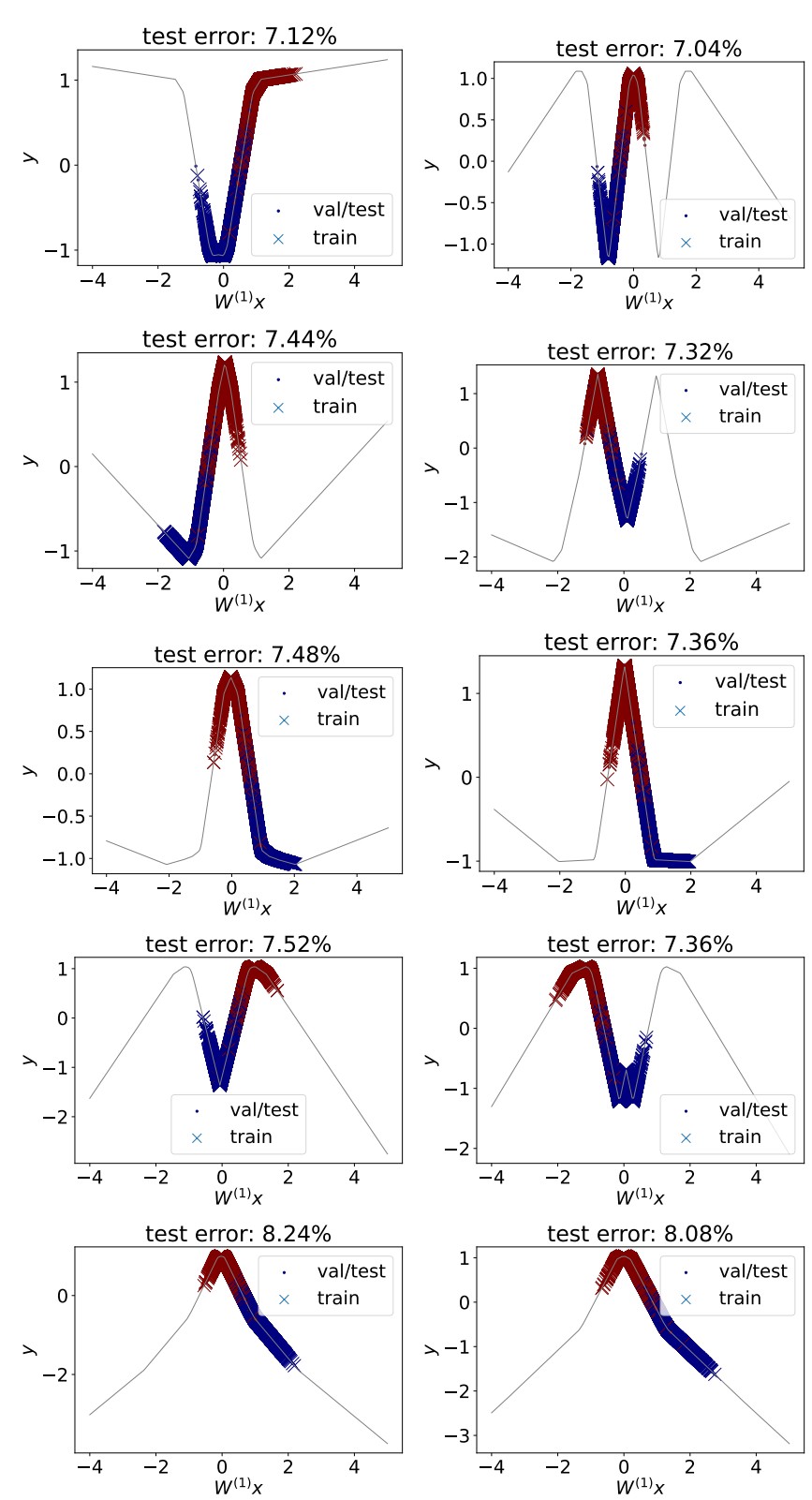

Figure 6: 3-layer (left column) and 4-layer (right column) deep narrow networks trained on OpenAI embeddings of IMDB reviews. Each row is a different training initialization.

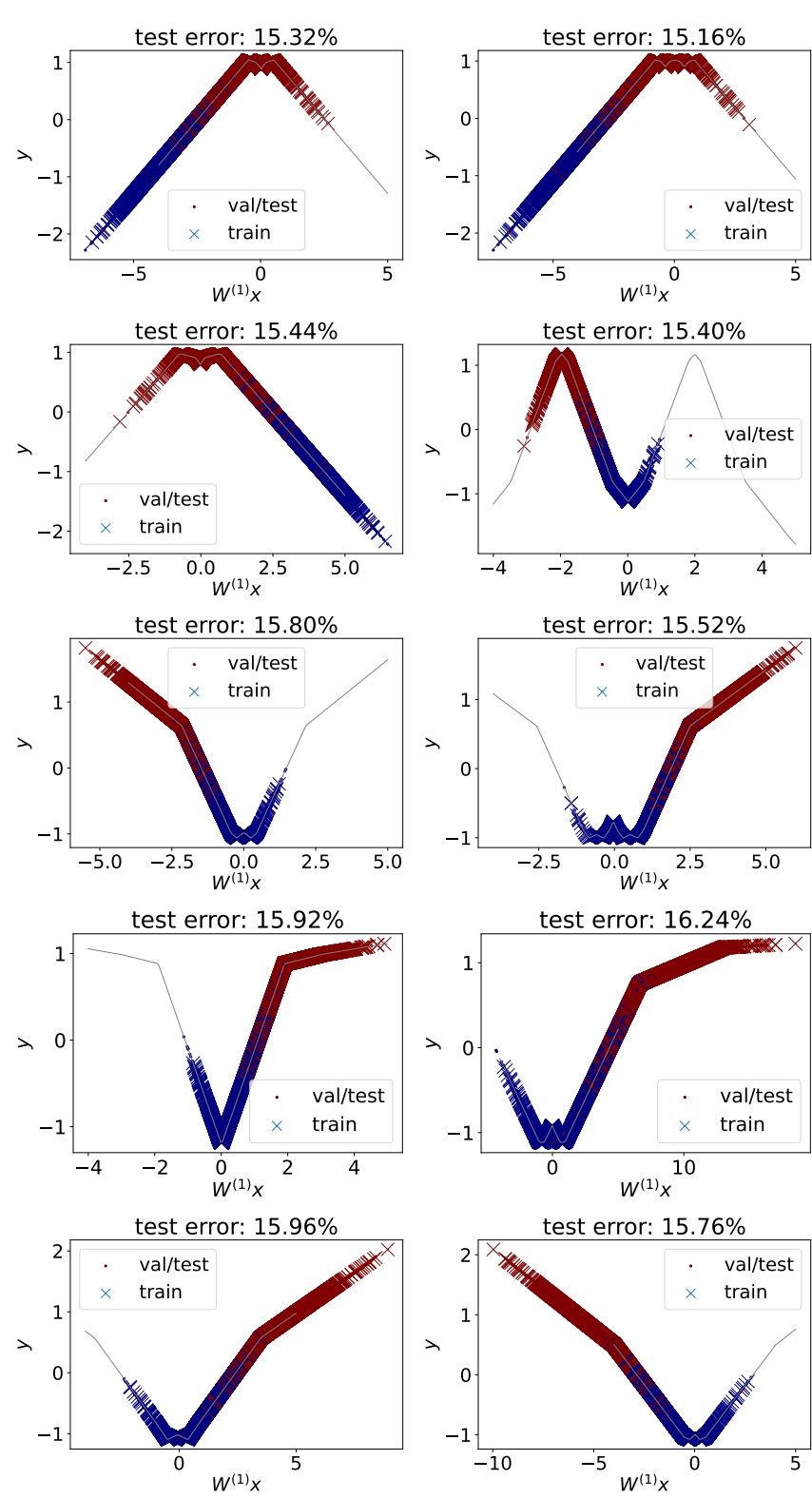

Figure 7: 3-layer (left column) and 4-layer (right column) deep narrow networks trained on Bert embeddings of IMDB reviews. Each row is a different training initialization.

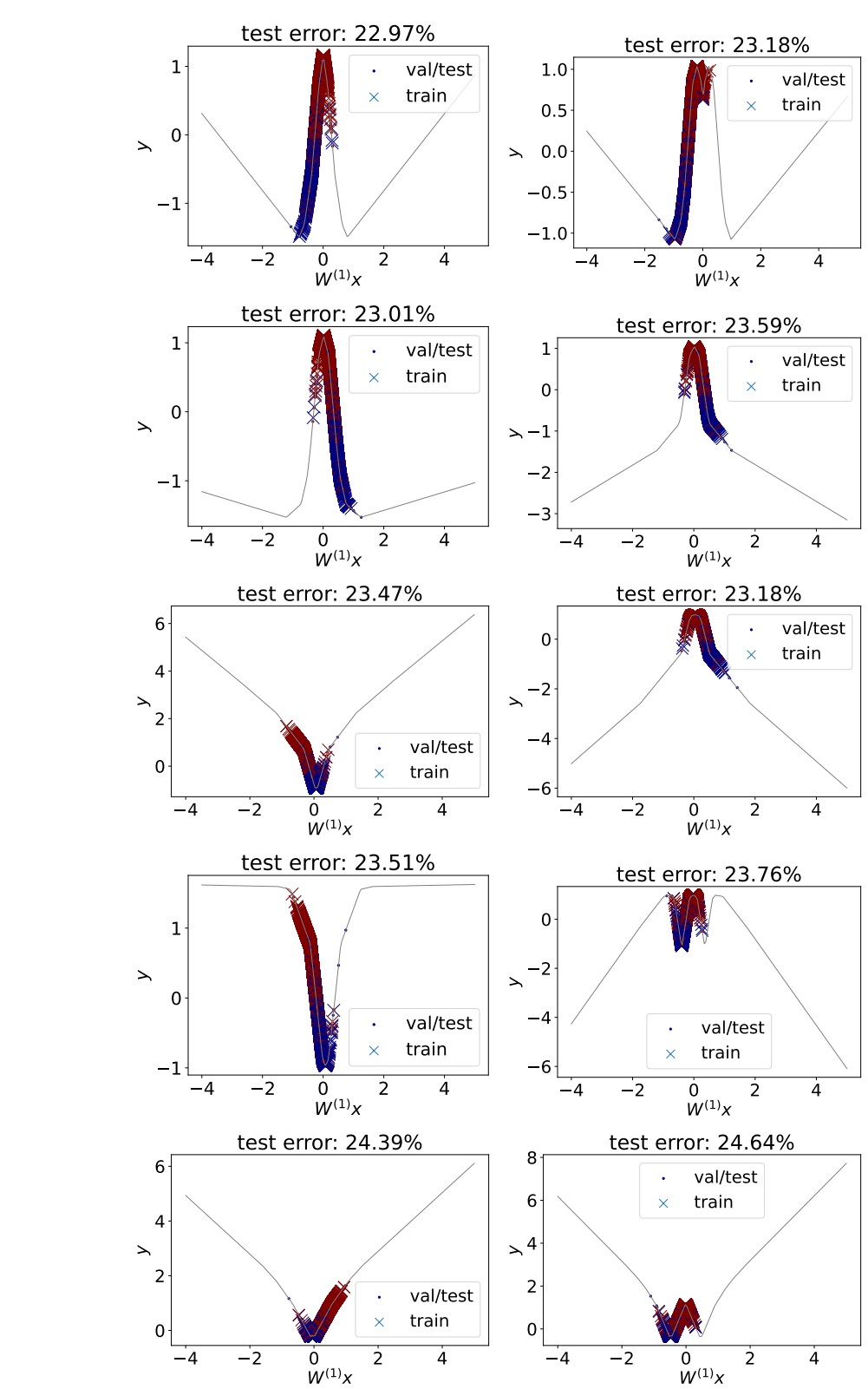

Figure 8: 3-layer (left column) and 4-layer (right column) deep narrow networks trained on OpenAI embeddings of GLUE-CoLA text. Each row is a different training initialization.

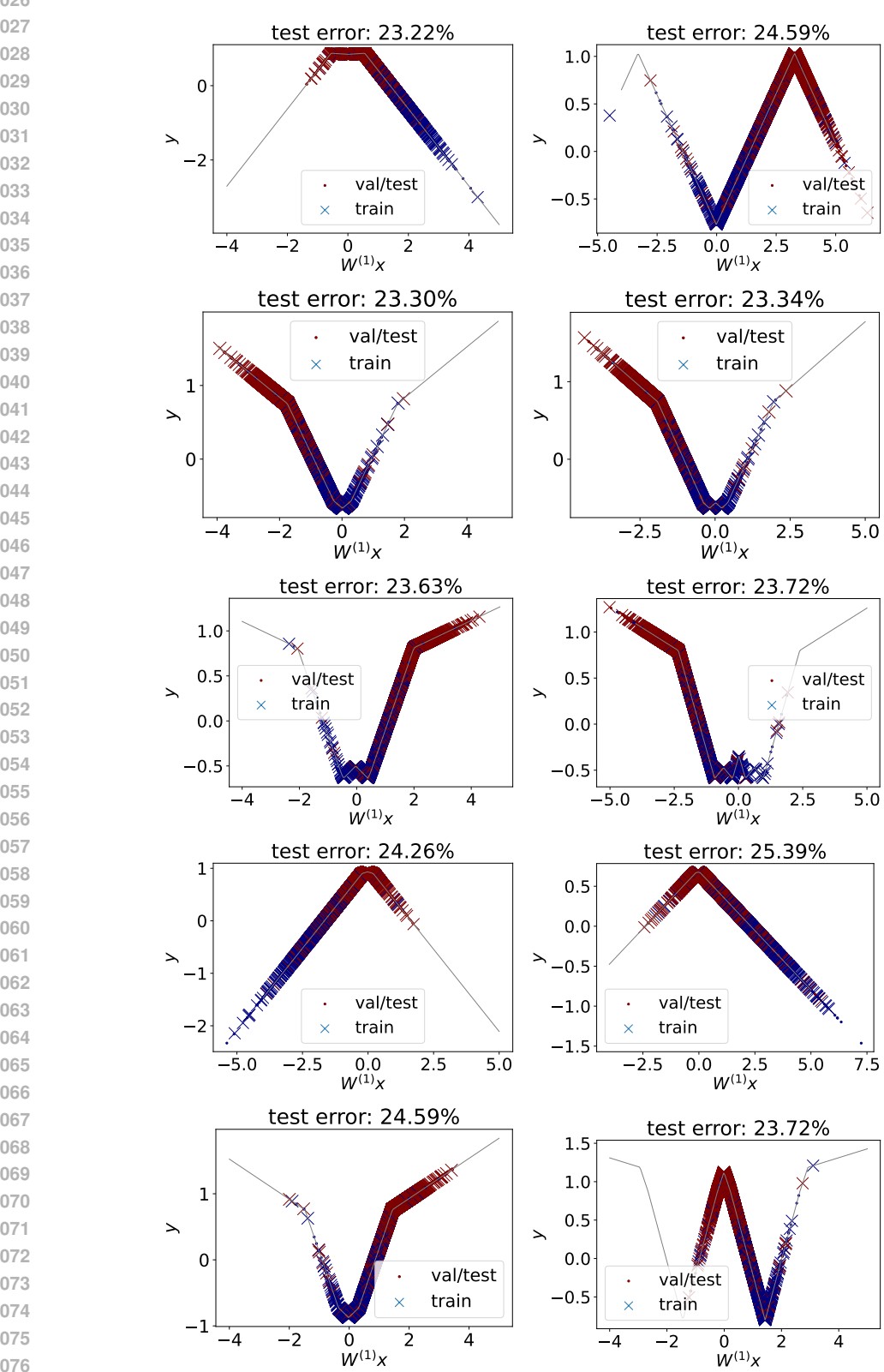

Figure 9: 3-layer (left column) and 4-layer (right column) deep narrow networks trained on Bert embeddings of GLUE-CoLA text. Each row is a different training initialization.

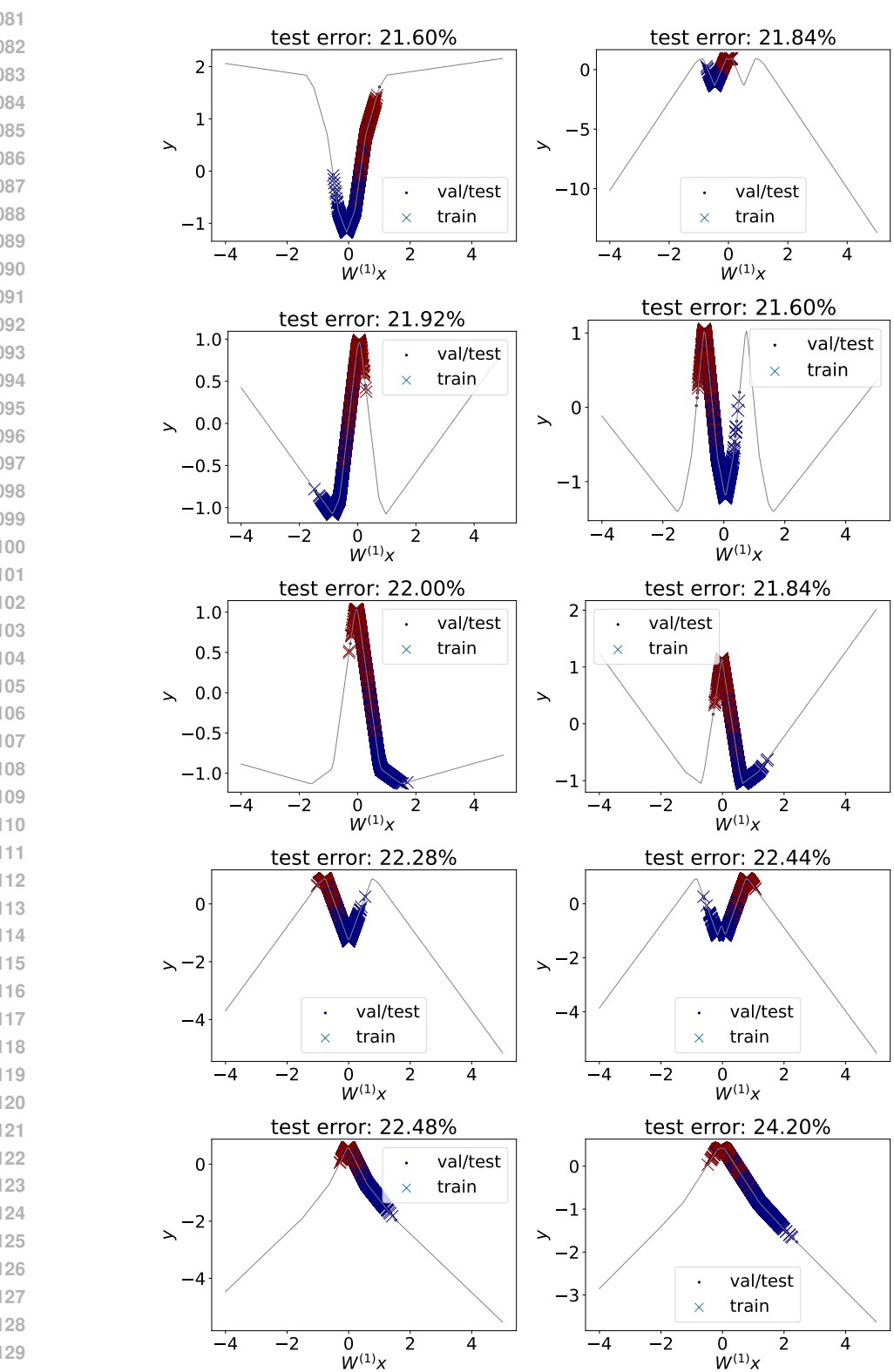

Figure 10: 3-layer (left column) and 4-layer (right column) deep narrow networks trained on OpenAI embeddings of GLUE-QQP text. Each row is a different training initialization.

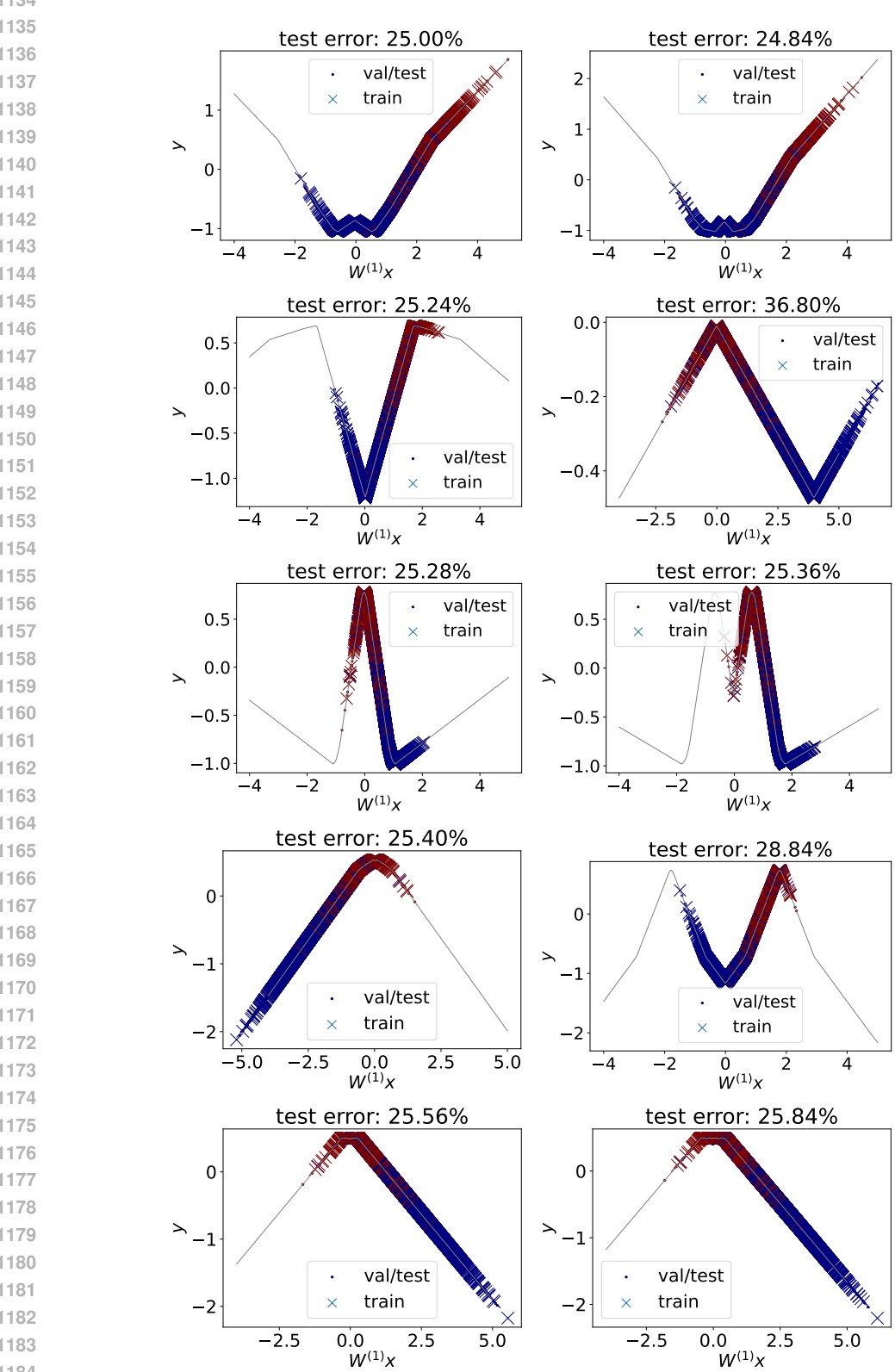

Figure 11: 3-layer (left column) and 4-layer (right column) deep narrow networks trained on Bert embeddings of GLUE-QQP text. Each row is a different training initialization.

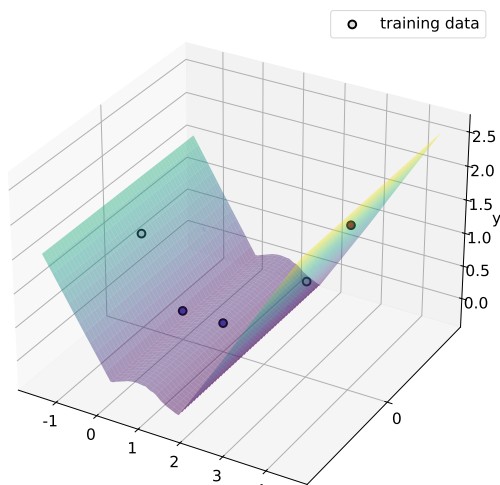

Figure 12: 3-layer ReLU neural net. A breakplane occurs at a reflection of training data, which is not in the training set.

$\left\{ \mathbf{W}^{(i,l)}, \alpha_i : i \in [m_L], l \in [L-1] \right\} \subset \theta$. All functions extend to vector and matrix-valued inputs elementwise.

The training problem (2) can be stated generally as

$$\min_{\theta \in \Theta} \mathcal{L}_{\mathbf{y}} \left( f_L \left( \mathbf{X}; \theta \right) \right) + \frac{\beta}{L} r(\theta_w),$$

where $\mathcal{L}_{\mathbf{y}} : \mathbb{R}^N \to \mathbb{R}$ is called the *loss* function that is convex and parameterized by $\mathbf{y}$, and $r(\theta)$ is a regularization term that penalizes large weight magnitudes. In (2), the loss function is $\mathcal{L}_{\mathbf{y}}(\cdot) = \frac{1}{2} \left\| \cdot - \mathbf{y} \right\|_2^2$. We can omit the external bias $\xi$ from the network $f_L \left( \mathbf{X}; \theta \right)$ by absorbing it into the loss $\mathcal{L}_{\mathbf{y}}$. We assume $r(\theta_w)$ is of the form

$$r(\theta_w) = \sum_{i=1}^{m_L} \left( \left( r^{(i,L)}(\alpha_i) \right)^L + \sum_{l=1}^{L-1} \left( r^{(i,l)} \left( \mathbf{W}^{(i,l)} \right) \right)^L \right) \tag{19}$$

where $r^{(i,l)}$ is a nonnegative function that is positively homogeneous, i.e., for any positive scalar $a$, $r^{(i,l)} \left( a\mathbf{W} \right) = ar^{(i,l)} \left( \mathbf{W} \right)$. In (2), the regularization is $r^{(i,l)}(x) = \|x\|_1$.

**Lemma D.1.** *The training problem is equivalent to the rescaled problem*

$$\min_{\theta \in \Theta : r^{(i,l)} \left( \mathbf{W}^{(i,l)} \right) = 1} \mathcal{L}_{\mathbf{y}} \left( f_L \left( \mathbf{X}; \theta \right) \right) + \beta \sum_{i=1}^{m_L} r^{(i,L)} \left( \alpha_i \right). \tag{20}$$

*Proof.* By the AM-GM inequality on (19), a lower bound on the training problem is

$$\min_{\theta \in \Theta} \mathcal{L}_{\mathbf{y}} \left( f_L \left( \mathbf{X}; \theta \right) \right) + \beta \sum_{i=1}^{m_L} r^{(i,L)}(\alpha_i) \prod_{l=1}^{L-1} r_{(i,l)} \left( \mathbf{W}^{(i,l)} \right). \tag{21}$$

Consider the minimization problem

$$\min_{\theta \in \Theta : r^{(i,l)} \left( \mathbf{W}^{(i,l)} \right) = 1} \mathcal{L}_y \left( f_L \left( \mathbf{X}; \theta \right) \right) + \beta \sum_{i=1}^{m_L} r^{(i,L)}(\alpha_i) \prod_{l=1}^{L-1} r_{(i,l)} \left( \mathbf{W}^{(i,l)} \right). \tag{22}$$

Problem (22) is an upper bound on (21). Given optimal $\left\{\mathbf{W}^{(i,l)}, \alpha_i\right\}$ in (21), the rescaled parameters $\mathbf{W}^{(i,l)'} = \mathbf{W}^{(i,l)}/r^{(i,l)}\left(\mathbf{W}^{(i,l)}\right)$ and $\alpha_i' = \alpha_i \prod_{l=1}^{L-1} r^{(i,l)}\left(\mathbf{W}^{(i,l)}\right)$ (and rescaled bias parameters) achieve the same objective in (22). Hence (22) and (21) are equivalent. Given an optimal parameter $q \in \left\{\mathbf{W}^{(i,l)}, \alpha_i\right\}$ in (22), the rescaled parameters $q' = |\alpha_i|^{\frac{1}{L}} q$ (and rescaled bias parameters) achieve the same objective in the training problem, which is therefore equivalent to (22). Simplifying (22) gives (20). $\qquad\square$

Let $\mathbf{X}^{(L)} = \mathbf{X}^{(1,L)}$. Assume $r^{(i,l)}(x) = \|x\|_1$.

**Lemma D.2.** *A lower bound on the rescaled training problem is*

$$\max_{\lambda \in \mathbb{R}^N} - \mathcal{L}_{\mathbf{y}}^*(\lambda) \quad s.t. \quad \max_{\theta \in \Theta} \left|\lambda^T \mathbf{X}^{(L)}\right| \leq \beta, \tag{23}$$

*where $f^*(\mathbf{x}) := \max_{\mathbf{x}} \left\{\mathbf{z}^T \mathbf{x} - f(\mathbf{x})\right\}$ is the convex conjugate of $f$.*

*Proof.* Find the dual of the rescaled training problem by rewriting it as

$$\min_{\theta \in \Theta} \mathcal{L}_{\mathbf{y}}(\mathbf{z}) + \beta \|\boldsymbol{\alpha}\|_1, \quad \text{s.t.} \quad \mathbf{z} = \sum_{i=1}^{m_L} \alpha_i \mathbf{X}^{(i,L)}. \tag{24}$$

The Lagrangian of problem (24) is $L(\lambda, \theta) = \mathcal{L}_{\mathbf{y}}(\mathbf{z}) + \beta \|\boldsymbol{\alpha}\|_1 - \lambda^T \mathbf{z} + \sum_{i=1}^{m_L} \lambda^T \mathbf{X}^{(i,L)} \alpha_i$. Minimize the Lagrangian over $\mathbf{z}$ and $\boldsymbol{\alpha}$ and use Fenchel duality (Boyd & Vandenberghe, 2004). The dual of (24) is

$$\max_{\lambda \in \mathbb{R}^N} - \mathcal{L}_{\mathbf{y}}^*(\lambda) \quad \text{s.t.} \quad \max_{\theta \in \Theta} \left|\lambda^T \mathbf{X}^{(i,L)}\right| \leq \beta, i \in [m_L]. \tag{25}$$

Observe $\mathbf{X}^{(i,L)}$ is of the same form for all $i \in [m_L]$. So the $m_L$ constraints in (25) collapse to just one constraint. Then we can write (25) as (23). $\qquad\square$

# E  DEEP NARROW NETWORK

Let $n^{(1)}, \cdots, n^{(L-1)} \in [N]$. Define the *breakplane sets* $\mathcal{K}^{(L)} = \{\mathbf{x}_{n^{(L-1)}}\}$ if $L = 2$ and $\mathcal{K}^{(L)} = \left\{\mathbf{X}_{n^{(L-2)}}^{(L-2)}, \frac{\mathbf{X}_{n^{(L-1)}}^{(L-2)} + \mathbf{X}_{n^{(L-2)}}^{(L-2)}}{2}\right\}$ otherwise. For $\mathbf{x}'_{j_1} \in \mathcal{K}^{(3)}$, let $\mathbf{M}_1 = \mathbf{X} - \mathbf{1}\mathbf{x}'_{j_1}$ and

$$f_{\mathbf{x}'_{j_1}} : \mathbb{R}^d \rightarrow \mathbb{R}^N, \qquad f_{\mathbf{x}'_{j_1}}(\mathbf{W}) = \mathbf{M}_1 \mathbf{W}$$
$$= \left(\mathbf{X} - \mathbf{1}\mathbf{x}'_{j_1}\right) \mathbf{W} \tag{26}$$
$$S_{\mathbf{x}'_{j_1}} : \mathbb{R}^d \rightarrow \{-1, 1\}^N, \quad S_{\mathbf{x}'_{j_1}}(\mathbf{W}) = \text{sign}\left(f_{\mathbf{x}'_{j_1}}(\mathbf{W})\right).$$

For $\mathbf{d}^{(1)} \in S_{\mathbf{x}'_{j_1}}\left(\mathbb{R}^d\right)$, let $\mathbf{M}_2 = \left(\text{Diag}(\mathbf{d}^{(1)}) - d_{n^{(2)}}^{(1)} \mathbf{E}_{n^{(2)}}\right) \mathbf{M}_1$ and

$$f_{\mathbf{d}^{(1)}, \mathbf{x}'_{j_1}, n^{(2)}} : \mathbb{R}^d \rightarrow \mathbb{R}^N, \qquad f_{\mathbf{d}^{(1)}, \mathbf{x}'_{j_1}, n^{(2)}}(\mathbf{W}) = \mathbf{M}_2 \mathbf{W}$$
$$= \left|\left(\mathbf{X}_n^{(1)} - \mathbf{x}'_{j_1}\right) \mathbf{W}\right| - \left|\left(\mathbf{X}_{n^{(2)}}^{(1)} - \mathbf{x}'_{j_1}\right) \mathbf{W}\right|$$
$$S_{\mathbf{d}^{(1)}, \mathbf{x}'_{j_1}, n^{(2)}} : \mathbb{R}^d \rightarrow \{-1, 1\}^{N \times N}, \quad S_{\mathbf{d}^{(1)}, \mathbf{x}'_{j_1}, n^{(2)}}(\mathbf{W}) = \text{sign}\left(f_{\mathbf{d}^{(1)}, \mathbf{x}'_{j_1}, n^{(2)}}(\mathbf{W})\right)$$
$$\tag{27}$$

where $\cdot$ denotes elementwise multiplication and $\mathbf{E}_{n^{(2)}}$ denotes a $N \times N$ matrix whose $(n^{(2)})^{\text{th}}$ column is $\mathbf{1}$ and all other elements are 0. These functions (26), (27) represent the sign patterns of a neuron in the first and second layers. Given $n^{(1)}, n^{(2)} \in [N], \mathbf{x}'_{j_1} \in \mathcal{K}^{(3)}$, a fixed sign pattern $\mathbf{d}^{(1)} \in S_{\mathbf{x}'_{j_1}}\left(\mathbb{R}^d\right)$, let $\mathbf{W}^{(1)} \in S_{\mathbf{x}'_{j_1}}^{-1}\left(\mathbf{d}^{(1)}\right)$ and

$$\mathcal{X}^{(3)} = \left\{\mathbf{x}_n - \mathbf{x}'_{j_1}\right\}_n \cup \left\{\mathbf{x}_n - \mathbf{x}_{n^{(2)}}\right\}_n \cup \left\{\mathbf{x}_n - R_{(\mathbf{x}_{n^{(2)}}, \mathbf{a})}\right\}_n \cup \{e_1, \cdots, e_d\}$$

$$\Delta\mathcal{X} = \left\{\pm \frac{\times_{i=1}^{d-1}\Delta\mathbf{x}_i}{\left\|\times_{i=1}^{d-1}\Delta\mathbf{x}_i\right\|_1} : \Delta\mathbf{x}_i \in \mathcal{X}^{(3)}\right\}. \tag{28}$$

Let $\delta B = \left\{\mathbf{W} \in \mathbb{R}^d : \|\mathbf{W}\|_1 = 1\right\}$ be the boundary of the $l_1$ ball of radius 1.

*Proof of Theorem 4.1.* In a deep narrow network (3), $\mathbf{W}^{(i,2)}, \cdots, \mathbf{W}^{(i,L-1)}, \mathbf{b}^{(l)}, \cdots, \mathbf{b}^{(L-1)}$ are scalar-valued and $\sigma(x) = |x|$. For fixed $\mathbf{W}^{(1)}, \cdots, \mathbf{W}^{(L-1)}$, $\mathbf{b}^{(l)} = \mathbf{b}^{(l)^*}$ is *optimal* if $\mathbf{b}^{(l)^*} \in \arg\max_{\mathbf{b}^{(l)}} \max_{\mathbf{b}^{(l+1)}} \cdots \max_{\mathbf{b}^{(L-1)}} \left|\sum_{n=1}^N \lambda_n \mathbf{X}_n^{(L)}\right|$, and analogously for $\mathbf{W}^{(l)}$. We will analyze the constraint of the dual (23):

$$\max_{\theta \in \Theta} \left|\lambda^T \mathbf{X}^{(L)}\right| \leq \beta. \tag{29}$$

Satisfying (29) requires $\mathbf{1}^T\lambda = 0$, otherwise $\max_{\theta \in \Theta}\left|\lambda^T\mathbf{X}^{(L)}\right| = \infty$. So assume $\mathbf{1}^T\lambda = 0$. We will use the following property: if $f : \mathbb{R} \to \mathbb{R}$ is a bounded piecewise linear function, $\arg\max_x |f(x)|$ contains a breakplane (or a *breakpoint* in $\mathbb{R}$) of $f$. The objective in the dual constraint (29) is $\left|\lambda^T\mathbf{X}^{(L)}\right| = \left|\sum_{n=1}^N \lambda_n \mathbf{X}_n^{(L)}\right|$ where

$$\mathbf{X}_n^{(L)} = \left|\mathbf{X}_n^{(L-1)}\mathbf{W}^{(L-1)} + \mathbf{b}^{(L-1)}\right|. \tag{30}$$

The breakplanes of $\mathbf{X}_n^{(L)}$ as a function of $\mathbf{b}^{(L-1)}$ occur at $\mathbf{b}^{(L-1)} = -\mathbf{X}_n^{(L-1)}\mathbf{W}^{(L-1)}$. Therefore for some $n^{(L-1)} \in [N]$, $\mathbf{b}^{(L-1)^*} = -\mathbf{X}_{n^{(L-1)}}^{(L-1)}\mathbf{W}^{(L-1)}$ is optimal. Now suppose $L > 2$. Plugging $\mathbf{b}^{(L-1)^*}$ into $\mathbf{X}_n^{(L)}$ (30) gives

$$\mathbf{X}_n^{(L)} = \left|\left(\mathbf{X}_n^{(L-1)} - \mathbf{X}_{n^{(L-1)}}^{(L-1)}\right)\mathbf{W}^{(L-1)}\right|$$
$$= \left|\left|\mathbf{X}_n^{(L-2)}\mathbf{W}^{(L-2)} + \mathbf{b}^{(L-2)}\right| - \left|\mathbf{X}_{n^{(L-1)}}^{(L-2)}\mathbf{W}^{(L-2)} + \mathbf{b}^{(L-2)}\right|\right|. \tag{31}$$

By (31), for fixed $n^{(L-1)}$, the set of breakplanes of $\sum_n \lambda_n \mathbf{X}_n^{(L)}$ as a function of $\mathbf{b}^{(L-2)}$ is $\left\{-\mathbf{x}'_{n^{(L-2)}}\mathbf{W}^{(L-2)} : \mathbf{x}'_{n^{(L-2)}} \in \mathcal{K}^{(L)}, n^{(L-2)} \in [N]\right\}$. So there exists $n^{(L-2)} \in [N]$ and $\mathbf{x}'_{n^{(L-2)}} \in \mathcal{K}^{(L)}$ such that

$$\mathbf{b}^{(L-2)^*} = -\mathbf{x}'_{n^{(L-2)}}\mathbf{W}^{(L-2)} \tag{32}$$

is optimal.

Now suppose $L = 3$. Let $\mathbf{x}'_{j_1} \in \mathcal{K}_{n^{(1)}, n^{(2)}}$. By (32), $\mathbf{b}^{(1)} = -\mathbf{x}'_{j_1}\mathbf{W}^{(1)}$ and plugging this into (31) gives

$$\mathbf{X}_n^{(3)} = \left|\left|\left(\mathbf{X}_n^{(1)} - \mathbf{x}'_{j_1}\right)\mathbf{W}^{(1)}\right| - \left|\left(\mathbf{X}_{n^{(2)}}^{(1)} - \mathbf{x}'_{j_1}\right)\mathbf{W}^{(1)}\right|\right| \tag{33}$$

By (33), the dual constraint (29) is equivalent to the following:

$$\max_{\mathbf{W}^{(2)} \in \{-1,1\}} \max_{n^{(1)}, n^{(2)} \in [N]} \max_{\mathbf{x}'_{j_1} \in \mathcal{K}^{(3)}} \max_{\mathbf{d}^{(1)} \in S_{\mathbf{x}'_{j_1}}(\mathbb{R}^d)} \max_{\mathbf{d}^{(2)} \in S_{\mathbf{d}^{(1)}, \mathbf{x}'_{j_1}, n^{(2)}}(\mathbb{R}^d)} \max_{\mathbf{W}^{(1)} \in S_{\mathbf{a}}^{-1}(\mathbf{d}^{(1)}) \cap S_{\mathbf{d}^{(1)}, \mathbf{x}'_{j_1}, n^{(2)}}^{-1}(\mathbf{d}^{(2)}) \cap \delta B} \left|\sum_{n=1}^N \lambda_n \mathbf{X}_n^{(L)}\right| \leq \beta,$$

$$\mathbf{1}^T\lambda = 0. \tag{34}$$

For each fixed $\mathbf{d}^{(1)} \in S_{\mathbf{x}'_{j_1}}(\mathbb{R}^d)$ and $\mathbf{d}^{(2)} \in S_{\mathbf{d}^{(1)}, \mathbf{x}'_{j_1}, n^{(2)}}(\mathbb{R}^d)$, the objective term $\mathbf{X}_n^{(L)} = \left(\left(\left(\mathbf{X}_n - \mathbf{x}'_{j_1}\right)\mathbf{W}^{(1)}\right)d_n^{(1)} - \left(\left(\mathbf{X}_{n^{(2)}} - \mathbf{x}'_{j_1}\right)\mathbf{W}^{(1)}\right)d_{n^{(2)}}^{(1)}\right)\mathbf{d}_n^{(2)}$ is linear in $\mathbf{W}^{(1)}$. Let

$B=\left\{\mathbf{W}\in\mathbb{R}^d:\|\mathbf{W}\|_1\leq 1\right\}$ be the filled $l_1$ ball of radius 1. If we replace $\delta B$ by $B$ in (34), the constraint (34) remains equivalent since the $\mathbf{W}^{(1)}$ that maximizes the left hand side of (34) will have the largest feasible magnitude. Then for each fixed $n^{(1)},n^{(2)},\mathbf{x}'_{j_1},\mathbf{b}^{(1)},\mathbf{d}^{(1)}$, the left hand side of the dual constraint (29) is the maximization of a linear function of $\mathbf{W}^{(1)}$ over the polytope

$$\mathcal{P}=S_{\mathbf{a}}^{-1}\left(\mathbf{d}^{(1)}\right)\cap S_{\mathbf{d}^{(1)},\mathbf{x}'_{j_1},n^{(2)}}^{-1}\left(\mathbf{d}^{(2)}\right)\cap B \tag{35}$$

which is equivalent to the same maximization over the finite set of extremal points of $\mathcal{P}$. Let $\mathbf{E}_{n^{(2)}}$ be a $N\times N$ matrix whose $(n^{(2)})^{\text{th}}$ column is $\mathbf{1}$ and all other elements are 0. The first-layer sign pattern function is the sign of a linear function of $\mathbf{W}$ : $\text{sign}\left(\mathbf{X}-\mathbf{1}\mathbf{x}'_{j_1}\right)$. The second-layer sign pattern function can also be written as the sign of a linear function of $\mathbf{W}$, as $S_{\mathbf{d},\mathbf{a},n}(\mathbf{W})=\text{sign}\left(\left(\text{Diag}(\mathbf{d})-d_n\mathbf{E}_n\right)\left(\mathbf{X}-\mathbf{1}\mathbf{x}'_{j_1}\right)\mathbf{W}\right)$, where $\cdot$ denotes elementwise multiplication. Therefore we can apply Lemma 19 in (Pilanci, 2023b) as follows. Let $\mathbf{I}$ denote an identity matrix, and let

$$\mathbf{M}=\begin{pmatrix}\mathbf{M}_1\\\mathbf{M}_2\\\mathbf{I}\end{pmatrix}.$$

The size of $\mathbf{M}$ is $(2N+d)\times d$. For $S\subset[N]$, let $\mathbf{M}_S$ denote a submatrix of $\mathbf{M}$ consisting of the rows indexed by $S$. The matrix $\mathbf{M}$ depends on $\mathbf{x}'_{j_1},\mathbf{d}^{(1)}$ and $n^{(2)}$. Let

$$\mathcal{E}_{\mathbf{d}^{(1)},\mathbf{x}'_{j_1},n^{(2)}}=\left\{\mathbf{W}^{(1)}\in\mathbb{R}^d:\exists S\subset[N],|S|=d-1,\text{ s.t. }\mathbf{M}_s\mathbf{W}^{(1)}=\mathbf{0},\text{rank}(\mathbf{M}_s)=d-1\right\}. \tag{36}$$

Let $\mathbf{M}_{si}$ denote the $i^{\text{th}}$ row of the submatrix $\mathbf{M}_s$. By the orthogonality property of generalized cross products (Section 2),

$$\mathcal{E}_{\mathbf{d}^{(1)},\mathbf{x}'_{j_1},n^{(2)}}\cap\delta B=\left\{\pm\frac{\times_{i=1}^{d-1}\mathbf{M}_{si}}{\left\|\times_{i=1}^{d-1}\mathbf{M}_{si}\right\|_1}:\text{rank}(\mathbf{M}_s)=d-1\right\}\subset\Delta\mathcal{X} \tag{37}$$

where $\Delta\mathcal{X}$ is defined in (28). By Lemma 19 in (Pilanci, 2023b), the set of extremal points of $\mathcal{P}$ is

$$\mathcal{E}_{\mathbf{d}^{(2)},\mathbf{d}^{(1)},\mathbf{a},n^{(2)}}=S_{\mathbf{x}'_{j_1}}^{-1}\left(\mathbf{d}^{(1)}\right)\cap S_{\mathbf{d}^{(1)},\mathbf{x}'_{j_1},n^{(2)}}^{-1}\left(\mathbf{d}^{(2)}\right)\cap\mathcal{E}_{\mathbf{d}^{(1)},\mathbf{x}'_{j_1},n^{(2)}}\cap\delta B. \tag{38}$$

Therefore for fixed $n^{(1)},n^{(2)},\mathbf{a}$, plugging (37) into (38) gives

$$\bigcup_{\mathbf{d}^{(1)}\in S_{\mathbf{a}}(\mathbb{R}^d),\mathbf{d}^{(2)}\in S_{\mathbf{d}^{(1)},\mathbf{x}'_{j_1},n^{(2)}}(\mathbb{R}^d)}\mathcal{E}_{\mathbf{d}^{(2)},\mathbf{d}^{(1)},\mathbf{a},n^{(2)}}\subset\Delta\mathcal{X}. \tag{39}$$

Plugging in (39) as a superset of extremal points of $\mathcal{P}$ into (34) shows that the dual constraint (29) is equivalent to

$$\max_{\mathbf{W}^{(2)}\in\{-1,1\}}\max_{n^{(1)},n^{(2)}\in[N]}\max_{\mathbf{x}'_{j_1}\in\mathcal{K}_{n^{(1)},n^{(2)}}}\max_{\mathbf{W}^{(1)}\in\Delta\mathcal{X}}\left|\sum_{n=1}^N\lambda_n\mathbf{X}_n^{(L)}\right|\leq\beta,\mathbf{1}^T\lambda=0. \tag{40}$$

Using (18), we can write (12) as $\mathbf{b}^{(l)}=-\mathbf{X}_{n^{(l)}}^{(l)}$ for $l>1$, where $\mathbf{W}^{(l')}=1$ for $l'>1$.

The dual of (40) is the Lasso problem (4) where $\mathbf{A}_{i,j}$ is determined by (33), or

$$\mathbf{A}_i=\mathbf{X}^{(3)}\left(\mathbf{X}\right). \tag{41}$$

In (41), $\mathbf{X}^{(3)}\left(\mathbf{X}\right)$ is the output of the network $\mathbf{X}^{(3)}$ with data matrix $\mathbf{X}$ as input, where $\mathbf{W}^{(2)}=1$ and $\mathbf{b}^{(l)}$ determined by (12) (with $\mathbf{x}'_{j_1}=\mathbf{a}$, i.e. which have $i^{th}$ element

$$\mathbf{A}_{i,j}=\left|\left|\left(\mathbf{x}_i-\mathbf{x}'_{j_1}\right)\mathbf{W}^{(1)}\right|-\left|\left(\mathbf{x}_{n^{(2)}}-\mathbf{x}'_{j_1}\right)\mathbf{W}^{(1)}\right|\right| \tag{42}$$

The Lasso problem is the bidual of the rescaled problem, and therefore a lower bound on it, which is equivalent to the original training problem. In fact, Remark E.3 shows that the Lasso problem is equivalent to the original training problem, and the reconstruction holds. Lastly, rename $j_0 = n^{(2)}, j_{-1} = n^{(1)}$. The volume formulation (9) holds from Remark B.1 and (16). The orthogonality property of $\mathbf{W}^{(1)}$ holds by the orthogonality property of cross products.

$\square$

*Proof of Lemma 4.2.* Since $j_{-1}, j_0, j_2, j_4, \cdots, j_{2(d-1)} \in [N]$, there are $N^{d+1}$ choices for determining $\mathbf{x}_{j_{-1}}, \mathbf{x}_{j_0}, \mathbf{x}_{j_2}, \mathbf{x}_{j_4}, \cdots, \mathbf{x}_{j_{2(d-1)}}$. For each $j_{-1}, j_0$ there are $\left\| \left\{ \frac{\mathbf{x}_{j_{-1}} + \mathbf{x}_{j_0}}{2}, \mathbf{x}_{j_{-1}} \right\} \right\| = 2$ options for $\mathbf{x}'_{j_1}$. For each $j_{-1}, j_0, \mathbf{x}_{j_{2k}}$, there are $\left| \left\{ \mathbf{x}'_{j_1}, R_{\left( \mathbf{x}_{j_0}, \mathbf{x}'_{j_1} \right)} \right\} \cup \{\mathbf{x}_{j_{2k}} - e_l\}_{l \in [d]} \right| = 2 + d$ options for $\mathbf{x}'_{j_{2k+1}}$. So the total number of options for $j$ is at most $2(N^{d+1})(2+d)^{d-1} = O((Nd)^d)$. The complexity for training a 2-layer network must be exponential in $d$ unless $\mathbf{P} = \mathbf{NP}$ Pilanci & Ergen (2020). $\square$

**Remark E.1.** *The 2-layer feature function is*

$$f(\mathbf{x}) = \left| (\mathbf{x} - \mathbf{x}_{n^{(1)}}) \overbrace{\frac{\times_{i=1}^{d-1} \Delta \mathbf{x}_i}{\left\| \times_{i=1}^{d-1} \Delta \mathbf{x}_i \right\|_1}}^{\mathbf{W}^{(1)} \perp \Delta \mathbf{x}_1, \cdots, \Delta \mathbf{x}_{d-1}} \right|. \tag{43}$$

*Similarly the 3-layer feature function is*

$$f(\mathbf{x}) = \left\| \left| \left( \mathbf{x} - \mathbf{x}'_{j_1} \right) \overbrace{\frac{\times_{i=1}^{d-1} \Delta \mathbf{x}_i}{\left\| \times_{i=1}^{d-1} \Delta \mathbf{x}_i \right\|_1}}^{\mathbf{W}^{(1)} \perp \Delta \mathbf{x}_1, \cdots, \Delta \mathbf{x}_{d-1}} \right| - \overbrace{\left| \left( \mathbf{x}_{n^{(2)}} - \mathbf{x}'_{j_1} \right) \frac{\times_{i=1}^{d-1} \Delta \mathbf{x}_i}{\left\| \times_{i=1}^{d-1} \Delta \mathbf{x}_i \right\|_1} \right|}^{-b^{(2)}} \right\|. \tag{44}$$

**Definition E.2.** *Given an optimal solution to the Lasso problem whose features are (11), the L-layer reconstruction is as follows. Set $\boldsymbol{\alpha} = \mathbf{z}^*, \gamma = \gamma^*, \mathbf{W}^{(i,l)} = 1$ for $l > 1$, and for each dictionary column $\mathbf{A}_i$, consider the corresponding $\mathbf{W}^{(1)}, \mathbf{b}^{(l)}$ and let $\mathbf{W}^{(i,1)} = \mathbf{W}^{(1)}$ and $\mathbf{b}^{(i,l)} = \mathbf{b}^{(l)}$. For data feature biases, use (12) for $\mathbf{b}^{(l)}$. For $L = 3$, $\mathbf{W}^{(1)} \in \Delta \mathcal{X}^{(3)}$ (28). This gives a rescaled network. Then, let $\gamma_i = |\alpha_i|^{\frac{1}{L}}$. Finally change variables as $q' = \text{sign}(q)\gamma_i$ for $q \in \left\{ \alpha_i, \mathbf{W}^{(i,l)} \right\}$ and $\mathbf{b}^{(i,l)'} = \mathbf{b}^{(i,l)} (\gamma_i)^l$ for the reconstructed network. For example, a reconstructed 3-layer neural network from a Lasso solution $\mathbf{z}^*, \xi^*$ is*

$$f_3(\mathbf{x}; \theta) = \sum_j (z_j^*)^{\frac{1}{3}} \left\| (z_j^*)^{\frac{1}{3}} \left| (z_j^*)^{\frac{1}{3}} \left( \mathbf{x} - \mathbf{x}'_{j_1}{}^{(j)} \right) \frac{\times_{i=1}^{d-1} \Delta \mathbf{x}_i^{(j)}}{\left\| \times_{i=1}^{d-1} \Delta \mathbf{x}_i^{(j)} \right\|_1} \right| - (z_j^*)^{\frac{1}{3}} \left| (z_j^*)^{\frac{1}{3}} \left( \mathbf{x}_{n^{(2)}}^{(j)} - \mathbf{x}'_{j_1}{}^{(j)} \right) \frac{\times_{i=1}^{d-1} \Delta \mathbf{x}_i^{(j)}}{\left\| \times_{i=1}^{d-1} \Delta \mathbf{x}_i^{(j)} \right\|_1} \right| \right\| + \xi^*. \tag{45}$$

*where the index $(j)$ indexes over all possible $\mathbf{W}^{(1)}, \mathbf{x}'_{j_1}, \mathbf{x}_{n^{(2)}}$ corresponding to the $j^{\text{th}}$ feature vector.*

**Remark E.3.** *The rescaled network in Definition E.2 achieves the same value in the rescaled problem as the Lasso optimal value. Definition E.2 then "un-scales" the weights in the rescaled network to give a reconstructed network that achieves the same objective in the original training problem as the optimal Lasso value. The rescaled network, reconstructed network and the function $\sum_i z_i^* f_{j_j}(\mathbf{x}) + \xi^*$ are all equivalent as functions, but have different interpretations of neuron weights.*

*Proof of Theorem 4.3.* The wedge product formulation follows from Theorem 4.3. To show the distance formulation, observe that the features are given by (42) as

$$f_j(\mathbf{x}) = \left\| \left| \left( \mathbf{x} - \mathbf{x}'_{j_1} \right) \mathbf{W}^{(1)} \right| - \left| \left( \mathbf{x}_{n^{(2)}} - \mathbf{x}'_{j_1} \right) \mathbf{W}^{(1)} \right| \right\| \tag{46}$$

with $\mathbf{W}^{(1)} = \frac{\mathbf{w}}{\|\mathbf{w}\|_1} \in \Delta\mathcal{X}^{(3)}$ (28), where $\mathbf{w} = \times_{i=1}^{d-1} \Delta\mathbf{x}_i^{(j)}$. Let $l_1$ and $l_2$ be the hyperplanes defined by $\left(\mathbf{x} - R_{\left(\mathbf{x}_{\mathbf{n}^{(2)}},\mathbf{x}'_{j_1}\right)}\right)\mathbf{w} = 0$ and $\left(\mathbf{x} - R_{\left(\mathbf{x}_{\mathbf{n}^{(2)}},\mathbf{x}'_{j_1}\right)}\right)\mathbf{w} = 0$, respectively. Let $l = l_1 \cup l_2$. By Remark B.2,

$$
\begin{aligned}
f_j(\mathbf{x}) &= \min\left\{\left|\left(\mathbf{x} - \mathbf{x}_{n^{(2)}}\right)\mathbf{W}^{(1)}\right|, \left|\left(\mathbf{x} - R_{\left(\mathbf{x}_{\mathbf{n}^{(2)}},\mathbf{x}'_{j_1}\right)}\right)\mathbf{W}^{(1)}\right|\right\} \\
&= r(\mathbf{w})\min\left\{\left|\left(\mathbf{x} - \mathbf{x}_{n^{(2)}}\right)\frac{\mathbf{w}}{\|\mathbf{w}\|_2}\right|, \left|\left(\mathbf{x} - R_{\left(\mathbf{x}_{\mathbf{n}^{(2)}},\mathbf{x}'_{j_1}\right)}\right)\frac{\mathbf{w}}{\|\mathbf{w}\|_2}\right|\right\} \qquad (47) \\
&= r(\mathbf{w})\min\{d(\mathbf{x},l_1), d(\mathbf{x},l_2)\} \\
&= r(\mathbf{w})d(\mathbf{x},l).
\end{aligned}
$$

Now rename $j_0 = n^{(2)}, j_{-1} = j_1$. Observe that if $\mathbf{x}'_{j_1} = \frac{\mathbf{x}_{j_{-1}} + \mathbf{x}_{j_0}}{2}$ then $R_{\left(\mathbf{x}_{j_0},\mathbf{x}'_{j_1}\right)} = \mathbf{x}_{\mathbf{x}_{j_{-1}}}$. A case analysis on $\mathbf{x}'_{j_1} \in \left\{\frac{\mathbf{x}_{j_{-1}} + \mathbf{x}_{j_0}}{2}, \mathbf{x}_{j_{-1}}\right\}$ gives (10). $\qquad\square$

*Proof of Theorem 4.4.* Recall the proof of Theorem 4.1 until (32). Continuing this argument of finding a finite set of possible breakplanes of $\lambda^T\mathbf{X}^{(L)}$ as a function of $\mathbf{b}^{(L-l)}$, plugging them in to $\mathbf{X}^{(L)}$ and further expanding $\lambda^T\mathbf{X}^{(L)}$ as a function of $\mathbf{b}^{(L-l-1)}$ until $l = L-2$ shows that for each fixed $\mathbf{W}^{(1)}$, there is a finite set of optimal $\mathbf{b}^{(1)}$, which determines a finite set of optimal $\mathbf{b}^{(2)}$, and so on determining a finite set of all other possible $\mathbf{b}^{(l)}$. We also see that for all $l > 1$, $\mathbf{W}^{(l)} = 1$ and $\mathbf{W}^{(l)} = -1$ are both optimal. Furthermore, the optimal values of $\mathbf{b}^{(l)}$ include

$$
\mathbf{b}^{(l)} = -\mathbf{X}_{n^{(L-l)}}^{(l)}\mathbf{W}^{(l)} \qquad (48)
$$

for $n^{(L-l)} \in [N]$. We can express $\mathbf{X}_n^{(L)}$ as a function of $\mathbf{W}^{(1)}$, generalizing (33). Next, partition $\mathbb{R}^d$ into a finite set of polytopes $\mathcal{P}_1, \cdots, \mathcal{P}_p$ corresponding to all possible sign patterns of activation arguments (generalizing (35)). The objective linearizes for $\mathbf{W}^{(1)}$ within each polytope, so maximizing $\left|\lambda^T\mathbf{X}^{(L)}\right|$ is equivalent to maximizing over the finite set of extreme points of the polytope, generalizing (37). So the dual constraint (29) consists of a finite set of linear constraints. As in the proof of Theorem 4.1, the bidual of the training problem is a Lasso problem whose dictionary columns are $\mathbf{X}^{(L)}(\mathbf{X})$ (generalizing (41)) over all possible optimal $\mathbf{W}^{(1)}, \mathbf{b}^{(l)}$. A similar reconstruction as Definition E.2 of an optimal neural network from a Lasso solution holds and shows that the Lasso problem and the training problem are equivalent. And (48) shows that the dictionary contains a data feature sub-library. Note that the 4-layer features plotted in the middle and right plots of Figure 2 depict $\mathbf{W}^{(1)}$ as orthogonal to the difference of training samples. By a similar argument as the proof of Theorem 4.1, this property holds for deeper layers as well. $\qquad\square$

## F   DEEP FEATURES WITH HIGHER ORDER REFLECTIONS

We define $\mathbf{x}$ to be an order-$l$ reflection of a *set* $\mathcal{S}^{(l)} = \{\mathbf{x}_0, \cdots, \mathbf{x}_k\}$ if $\mathbf{x}$ can be written recursively in the form $R_l = R_l(\mathcal{S}^{(l)}) \in \left\{R_{\left(R_{l-1}(\mathcal{S}^{(l-1)}),\mathbf{x}'\right)}, R_{\left(\mathbf{x}',R_{l-1}(\mathcal{S}^{(l)})\right)}\right\}$ where $\mathcal{S}^{(l-1)} \subset \mathcal{S}^{(l)}$ with possibly repeating elements and $\left|\mathcal{S}^{(l)}\right| = l+1$. Note that an order-$k$ reflection is also an order-$j$ reflection for any $j \geq k$. In the following lemma, let $a^{(L)}, b^{(L)} \in \mathbb{R}, \mathbf{c}^{(L)} \in \mathbb{R}^N, j_1, \cdots, j_L \in [N]$. For $l \in [L-1]$, recursively define $a^{(l)} = \left|a^{(l+1)} - c_{j_{l+1}}^{(l+1)}\right|, b^{(l)} = \left|b^{(l+1)} - c_{j_{l+1}}^{(l+1)}\right|, c_{j_l}^{(l)} = \left|c_{j_l}^{(l+1)} - c_{j_{l+1}}^{(l+1)}\right|$. Let $\mathcal{S}^{(l)} = \left\{a^{(l)}, b^{(l)}\right\} \cup \left\{c_{l'}^{(l)} : l' \leq l\right\}$. We will see that this recursively models the structure of a feature function in a data feature sub-library.

**Lemma F.1.** *Let $l \in [L]$. Let $R_0^{(l,+)} = R_0^{(l,-)} = R_{\left(a^{(l)},b^{(l)}\right)}$. For $l' \in \{0, \cdots, L-l-1\}$, there exist $l+l'+1$-order reflections $R_{2(l'+1)}^{(l+l'+1,+)}, R_{2(l'+1)}^{(l+l'+1,-)}$ of $\mathcal{S}^{(l+l'+1)}$ such that*

$$
R_{2l'}^{(l+l',\pm)} \in \left\{R_{2(l'+1)}^{(l+l'+1,+)} - c_{l+l'+1}^{(l+l'+1)}, c_{l+l'+1}^{(l+l'+1)} - R_{2(l'+1)}^{(l+l'+1,-)}\right\}. \qquad (49)
$$

*for $R_{2l'}^{(l+l',\pm)} \in \left\{R_{2l'}^{(l+l',+)}, R_{2l'}^{(l+l',-)}\right\}$.*

*Proof of Lemma F.1.* Observe that for any $l$,

$$R_{\left(a^{(l)},b^{(l)}\right)} = R_{\left(\left|a^{(l+1)}-c_{j_{l+1}}^{(l+1)}\right|,\left|b^{(l+1)}-c_{j_{l+1}}^{(l+1)}\right|\right)}$$

$$\in \pm \left\{ R_{\left(a^{(l+1)}-c_{j_{l+1}}^{(l+1)},b^{(l+1)}-c_{j_{l+1}}^{(l+1)}\right)}, R_{\left(a^{(l+1)}-c_{j_{l+1}}^{(l+1)},c_{j_{l+1}}^{(l+1)}-b^{(l+1)}\right)} \right\}$$

$$= \pm \left\{ -a^{(l+1)}+2b^{(l+1)}-c_{j_{l+1}}^{(l+1)}, -a^{(l+1)}-2b^{(l+1)}+3c^{(l+1)} \right\}$$

$$= \pm \left\{ R_{\left(a^{(l+1)},b^{(l+1)}\right)}-c_{j_{l+1}}^{(l+1)} = c_{j_{l+1}}^{(l+1)} - R_{\left(R_{\left(a^{(l+1)},b^{(l+1)}\right)},c_{j_{l+1}}^{(l+1)}\right)}, \right.$$

$$\left. R_{\left(a^{(l+1)},R_{\left(b^{(l+1)},c_{j_{l+1}}^{(l+1)}\right)}\right)}-c_{j_{l+1}}^{(l+1)} = c_{j_{l+1}}^{(l+1)} - R_{\left(R_{\left(a^{(l+1)},c_{j_{l+1}}^{(l+1)}\right)},b^{(l+1)}\right)} \right\}. \tag{50}$$

Therefore for some $a^{(l+1)'}, b^{(l+1)'}, c^{(l+1)'} \in \left\{ a^{(l+1)}, b^{(l+1)}, c_{j_{l+1}}^{(l+1)} \right\}$ and

$$R_2^{(l+1,+)}, R_2^{(l+1,-)} \in \left\{ R_{\left(R_{\left(a^{(l+1)'},b^{(l+1)'}\right)},c^{(l+1)'}\right)}, R_{\left(c^{(l+1)'},R_{\left(a^{(l+1)'},b^{(l+1)'}\right)}\right)} \right\} \tag{51}$$

which are order-2 reflections of $\left\{ a^{(l+1)}, b^{(l+1)}, c_{j_{l+1}}^{(l+1)} \right\}$, we have

$$R_{\left(a^{(l)},b^{(l)}\right)} = \left\{ c_{j_{l+1}}^{(l+1)} - R_2^{(l+1,-)} = R_2^{(l+1,+)} - c_{j_{l+1}}^{(l+1)} \right\}. \tag{52}$$

Applying (52) for $l' = l+1$ gives

$$R_{\left(a^{(l+1)},b^{(l+1)}\right)} \in \left\{ c_{j_{l+2}}^{(l+2)} - R_2^{(l+2,-)} = R_2^{(l+2,+)} - c_{j_{l+2}}^{(l+2)} \right\} \tag{53}$$

for some $R_2^{(l+2,+)}, R_2^{(l+2,-)}$ that are order-2 reflections of $\left\{ a^{(l+2)}, b^{(l+2)}, c_{j_{l+2}}^{(l+2)} \right\}$. Plugging (53) into (51) shows that for $R_2^{(l+1,\pm)} \in \left\{ R_2^{(l+1,+)}, R_2^{(l+1,-)} \right\}$ we have

$$R_2^{(l+1,\pm)} = R_{\left(R_{\left(a^{(l+1)'},b^{(l+1)'}\right)},c^{(l+1)'}\right)}$$

$$= 2\left| c^{(l+2)'} - c_{j_{l+2}}^{(l+2)} \right| - \left\{ c_{j_{l+2}}^{(l+2)} - R_2^{(l+2,-)} = R_2^{(l+2,+)} - c_{j_{l+2}}^{(l+2)} \right\}$$

$$\in \left\{ -c_{j_{l+2}}^{(l+2)} + 2c^{(l+2)'} - R_2^{(l+2,+)}, c_{j_{l+2}}^{(l+2)} - 2c^{(l+2)'} + R_2^{(l+2,-)} \right\}$$

$$= \left\{ R_{\left(R_2^{(l+2,+)},c^{(l+2)'}\right)} - c_{j_{l+2}}^{(l+2)} = c_{j_{l+2}}^{(l+2)} - R_{\left(R_{\left(R_2^{(l+2,+)},c^{(l+2)'}\right)},c_{j_{l+2}}^{(l+2)}\right)}, \right.$$

$$\left. = c_{j_{l+2}}^{(l+2)} - R_{\left(R_2^{(l+2,-)},c^{(l+2)'}\right)} = R_{\left(R_{\left(R_2^{(l+2,-)},c^{(l+2)'}\right)},c_{j_{l+2}}^{(l+2)}\right)} - c_{j_{l+2}}^{(l+2)} \right\} \tag{54}$$

or alternatively,

$$R_2^{(l+1,\pm)} = R_{\left(c^{(l+1)'}, R_{\left(a^{(l+1)'}, b^{(l+1)'}\right)}\right)}$$

$$= 2\left\{c_{j_{l+2}}^{(l+2)} - R_2^{(l+2,-)} = R_2^{(l+2,+)} - c_{j_{l+2}}^{(l+2)}\right\} - \left|c^{(l+2)'} - c_{j_{l+2}}^{(l+2)}\right|$$

$$\in \left\{-c_{j_{l+2}}^{(l+2)} + 2R_2^{(l+2,+)} - c^{(l+2)'}, c_{j_{l+2}}^{(l+2)} - 2R_2^{(l+2,-)} + c^{(l+2)'}\right\}$$

$$= \left\{ R_{\left(c^{(l+2)'}, R_2^{(l+2,+)}\right)} - c_{j_{l+2}}^{(l+2)} = c_{j_{l+2}}^{(l+2)} - R_{\left(R_{\left(c^{(l+2)'}, R_2^{(l+2,+)}\right)}, c_{j_{l+2}}^{(l+2)}\right)}, \right.$$

$$\left. c_{j_{l+2}}^{(l+2)} - R_{\left(c^{(l+2)'}, R_2^{(l+2,-)}\right)} = R_{\left(R_{\left(c^{(l+2)'}, R_2^{(l+2,-)}\right)}, c_{j_{l+2}}^{(l+2)}\right)} - c_{j_{l+2}}^{(l+2)}\right\} \tag{55}$$

for some $c^{(l+2)'} \in \left\{a^{(l+2)}, b^{(l+2)}, c_{j_{l+1}}^{(l+2)}\right\}$. Therefore (54) and (55) show that in all cases,

$$R_2^{(l+1,\pm)} \in \left\{R_4^{(l+2,+)} - c_{j_{l+2}}^{(l+2)}, c_{j_{l+2}}^{(l+2)} - R_4^{(l+2,-)}\right\} \tag{56}$$

for some order-4 reflections $R_4^{(l+2,+)}, R_4^{(l+2,-)}$ of $\left\{a^{(l+2)}, b^{(l+2)}, c_{j_{l+1}}^{(l+2)}, c_{j_{l+2}}^{(l+2)}\right\}$.

Applying a similar argument used to reach (56) for $l' = l + 2$ shows that

$$R_2^{(l+2,\pm)} \in \left\{R_4^{(l+3,+)} - c_{j_{l+3}}^{(l+3)}, c_{j_{l+3}}^{(l+3)} - R_4^{(l+3,-)}\right\} \tag{57}$$

for some order-4 reflections $R_4^{(l+3,+)}, R_4^{(l+3,-)}$ of $a^{(l+3)}, b^{(l+3)}, c_{j_{l+2}}^{(l+3)}, c_{j_{l+3}}^{(l+3)}$. Now plug (57) into the last lines of (54) and (55). Applying the same argument as (54) and (55) but changing the reflection order subscripts from 1 to 2 and from 2 to 4, and adding 1 to the layer superscripts $l + 1, l + 2$ shows that

$$R_4^{(l+2,\pm)} \in \left\{R_6^{(l+3,+)} - c_{j_{l+3}}^{(l+3)}, c_{j_{l+3}}^{(l+3)} - R_6^{(l+3,-)}\right\} \tag{58}$$

for some order-6 reflections $R_6^{(l+3,+)}, R_6^{(l+3,-)}$ of $\left\{a^{(l+3)}, b^{(l+3)}, c_{j_{l+1}}^{(l+3)}, c_{j_{l+2}}^{(l+3)}, c_{j_{l+3}}^{(l+3)}\right\}$.

Comparing (52), (56), (58) and repeating the same argument shows that

$$R_1^l = R_{\left(a^{(l)}, b^{(l)}\right)}$$

$$\in \left\{R_2^{(l+1,+)} - c^{(l+1)} = c^{(l+1)} - R_2^{(l+1,-)}\right\}$$

$$R_2^{(l+1,\pm)} \in \left\{R_4^{(l+2,+)} - c^{(l+2)}, c^{(l+2)} - R_4^{(l+2,-)}\right\}$$

$$R_4^{(l+2,\pm)} \in \left\{R_6^{(l+3,+)} - c^{(l+3)}, c^{(l+3)} - R_6^{(l+3,-)}\right\} \tag{59}$$

$$\vdots$$

$$R_{2l'}^{(l+l',\pm)} \in \left\{R_{(2(l'+1))}^{(l+l'+1,+)} - c^{(l+l'+1)}, c^{(l+l'+1)} - R_{(2(l'+1))}^{(l+l'+1,-)}\right\}.$$

This completes the proof. $\qquad\square$

**Proof sketch of Theorem 4.5:** We can easily show that 2 and 3-layer networks can be written using reflections. We extend this to deeper layers by first showing that reflections of $\mathbf{X}^{(l)}$ can be expressed as higher order reflections of $\mathbf{X}^{(k)}$ for $k < l$ (Lemma F.1). Then, we recursively plug the higher-order reflections into the deeper networks (63). This will show that for $l > 2$, the neural net is

$\mathbf{X}^{(L)}=\left|\mathbf{X}^{(L-l)}-R_{2(l-3)}^{(L-l)}\right|$ for some order-$2(l-3)$ reflection of the parallel units (features) at layer $L-l$. Finally, plug in $l=L-1$.

*Proof of Theorem 4.5.* The features $f_j(\mathbf{x})$ (11) of the data feature sub-library have bias parameters (12). In other words, $f_j(\mathbf{x})=\mathbf{X}^{(L)}(\mathbf{x})$ where for all $l\in[L-1]$ there exists $j_l\in[N]$ such that for all $\mathbf{x}\in\mathbb{R}^d$,

$$\mathbf{X}^{(l+1)}(\mathbf{x}) = \left|\mathbf{X}^{(l)}(\mathbf{x})-\mathbf{X}^{(l)}(\mathbf{x}_{j_l})\right|. \tag{60}$$

For shorthand, denote $\mathbf{X}^{(l)}=\mathbf{X}^{(l)}(\mathbf{x}), \mathbf{X}_j^{(l)}=\mathbf{X}^{(l)}(\mathbf{x}_{j_l})$. Then

$$\begin{aligned}
\mathbf{X}^{(L)} &= \left|\mathbf{X}^{(L-1)} - \mathbf{X}_{j_{L-1}}^{(L-1)}\right| \\
&= \left|\left|\mathbf{X}^{(L-2)} - \mathbf{X}_{j_{L-2}}^{(L-2)}\right| - \left|\mathbf{X}_{j_{L-1}}^{(L-2)} - \mathbf{X}_{j_{L-2}}^{(L-2)}\right|\right| \\
&\in \left\{\left|\mathbf{X}^{(L-2)} - \mathbf{X}_{j_{L-2}}^{(L-2)}\right|, \left|\mathbf{X}^{(L-2)} - R_{\left(\mathbf{x}_{j_{L-1}}^{(L-2)},\mathbf{x}_{j_{L-2}}^{(L-2)}\right)}\right|\right\}.
\end{aligned} \tag{61}$$

Therefore

$$\mathbf{X}^{(L)} = \left|\mathbf{X}^{(L-2)} - R_1^{(L-2)}\right| \tag{62}$$

for some order-1 reflection $R_1^{(L-2)}\in\left\{R_{\left(\mathbf{x}_{j_{L-2}}^{(L-2)},\mathbf{x}_{j_{L-2}}^{(L-2)}\right)}, R_{\left(\mathbf{x}_{j_{L-1}}^{(L-2)},\mathbf{x}_{j_{L-2}}^{(L-2)}\right)}\right\}$. Expand $\mathbf{X}^{(L-2)}$ in (62) with (60) and apply Lemma F.1 to $R_1^{(L-2)}$. Specifically, we set $a^{(l)},b^{(l)}\in\left\{\mathbf{X}_{j_{L-1}}^{(L-2)},\mathbf{X}_{j_{L-2}}^{(L-2)}\right\}$, let $j_a,j_b$ be the subscripts ($j_{L-1}$ or $j_{L-2}$) of $a^{(l)},b^{(l)}$, and let $a^{(l+1)}=\mathbf{X}_{j_a}^{(L-3)},b^{(l+1)}=\mathbf{X}_{j_b}^{(L-3)},c^{(l+1)}=\mathbf{X}_{j_{L-3}}^{(L-3)}$ in (52). This yields

$$\begin{aligned}
\mathbf{X}^{(L)} &= \left|\left|\mathbf{X}^{(L-3)} - \mathbf{X}_{j_{L-3}}^{(L-3)}\right| - \left\{R_2^{(L-3,+)} - \mathbf{X}_{j_{L-3}}^{(L-3)} = \mathbf{X}_{j_{L-3}}^{(L-3)} - R_2^{(L-3,-)}\right\}\right| \\
&= \left|\mathbf{X}^{(L-3)} - R_2^{(L-3,\pm)}\right|.
\end{aligned} \tag{63}$$

Next, repeat the same argument (63) but use (57) instead of (52) to get $\mathbf{X}^{(L)}=\left|\mathbf{X}^{(L-4)} - R_2^{(L-4,\pm)}\right|$. Continue repeating the same argument by using (49) for general $l$. Specifically, for fixed $j_a, j_b \in [N]$, for all $l \in [L]$, let $a^{(l)}=\mathbf{X}_{j_a}^{(L-l)}, b^{(l)}=\mathbf{X}_{j_a}^{(L-l)}$ and $c_{j_l}^{(l)}=\mathbf{X}_{j_{L-l}}^{(L-l)}$ in (57). We arrive at $\mathbf{X}^{(L)} = \left|\mathbf{X} - R_{2(L-3)}\right|$. Thus for $L > 3$, the neural net has breakplanes at reflections of training data of up to order-$2(L-3)$. The network has breakplanes at reflections of order 0 for $L = 2$ and order 1 for $L = 3$ (62), respectively. $\square$

