# OpenReview forum: "Black Boxes and Looking Glasses: Multilevel Symmetries, Reflection Planes, and Convex Optimization in Deep Networks"
_ICLR.cc/2025/Conference — ICLR 2025 Conference Withdrawn Submission_

### Official Review · Reviewer_1nMH · 2024-10-30

**Soundness:** 3
**Presentation:** 2
**Contribution:** 2
**Rating:** 5
**Confidence:** 2

**Summary:**

This work studies a specific class of "narrow" deep neural networks that are sums of a number of 1-neuron-L-layer networks. It shows a reduction from training such networks to a Lasso optimization problem, which importantly is convex and thus is much easier to find the global optimum. On top of the previous literature, this work considers a different activation function--absolute value--instead of ReLU. It reveals certain geometric properties of those networks related to reflection hyperplanes as a result of the choice of the activation.

**Strengths:**

- This work is a nice addition to the existing literature.
- It is nice that this work gives explicit constructions of the dictionary matrices of the Lasso problem, so that one can solve the convex problem as an alternative.
- The discovery of the reflection hyperplanes is interesting.

**Weaknesses:**

- The definition of the deep networks is a bit confusing since people rarely define it to be a sum of M copies of fully-connected nets. But this confusion is minor since this work then focuses on the "narrow" case which is basically a sum of M 1-neuron nets.
- But this architecture and this activation is far away from what is generally used.
- Also, even though this work presents nice connections between the training of the narrow nets and a convex problem, the tools and techniques seem to be very specialized to this setting. It is unclear whether they generalize to settings that are closer to what is used in practice.
- $x'$ is never specified around Theorem 3.2.
- In Theorem 4.1, the elements are defined by $A_{i,j}=f_j(x_i)$, but the role of $i$ is not specified in the definition of $f_j(x)$.
- The message behind the results is potentially hard to follow for people who have worked on learning theory but are unfamiliar with geometric algebra. It would be nice to give more emphasis on the message such as the end of Section 4.1.

**Questions:**

- In Theorem 4.1, what does it mean to have $x'\in S$ with some set $S$, whereas it should be a vector (to match the operations in $f$)? What would be its geometric interpretation?
- Could the authors define $x'$ before Theorem 3.2?
- Since the reflection hyperplanes seem to be a consequence of the choice of activation, and the authors interpret them as "concepts", I am curious what are the counterparts for other activations, say ReLU which is almost "half" of the absolute?

---

### Official Review · Reviewer_iMsG · 2024-11-01

**Soundness:** 1
**Presentation:** 2
**Contribution:** 2
**Rating:** 3
**Confidence:** 2

**Summary:**

The paper shows an equivalence between a specific class of neural networks (i.e. deep narrow network) and Lasso regression. This equivalence, together with a geometric algebra view, reveal geometric structures in the Lasso features.
The authors shows explicit Lasso features for a 3-layer deep narrow network, and use them to interpret network inference as measuring distances to planes, capturing reflections and symmetries of training data.

**Strengths:**

The paper adopt an interesting view of neural network. They embrace the tools of geometric algebra and use them to show explicitly the learned features in a training setting. This shred light on how networks "learn" and specifically on the importance of symmetries.

The idea of "concept" introduced in lines 381-386, although informal, nicely clarify an aspect of how predictions are made using the learned features.

Also the idea of "sparsity factor" is very interesting.

**Weaknesses:**

First of all I want to emphasize that I'm not very confident with geometric algebra, and this is probably an important reason why I struggle following the derivations. Having said that, I think the paper fails in being enough clear and formal.

For example, equation 12 is completely impossible to understand for me, the notation is totally off. What are $n^{(1)},\dots,n^{(L-1)}$? You define $n^{(1)}=x_{j_1}$  and then you refer to $x_{n^{(l)}}$, how can these be consistent?
Besides this example, which is probably a typo that I can't resolve, I think over the whole paper the notation is extremely hard to understand and not enough work has been done to make it accessible. (but again, it can very much only be me that I don't know the field enough)

The main concern I have with the paper is about the "narrow" networks, which indeed are a much simpler variant of standard networks. They are equivalent in the 2-layers case, but they are definitely not in the deeper case. For this reason I think the authors claim "we prove an equivalence between neural networks and Lasso problem" is not fair.

Moreover, I think the experiments are extremely limited. Beside the synthetic data ones, the one "using LLM embeddings" in my opinion are not convincing at all. The task you are solving is bi-classifying the embedding generated by a pre-trained LLM. I'm quite confident that if the LLM is trained well enough, than even a simple linear regression would manage to solve your task.

**Questions:**

Is there any additional capacity in a 3-layers narrow network over a 2-layers standard one?

What's the improvement over the result by (Pilanci, 2023b)?

Can the authors explain equation 12 notation?

What are the axes in Figure 3?

---

### Official Review · Reviewer_zeLX · 2024-11-01

**Soundness:** 3
**Presentation:** 2
**Contribution:** 2
**Rating:** 5
**Confidence:** 3

**Summary:**

The authors study the task of finding the optimal parameters of a special (albeit unconventional) form of neural network for a regression task. They show theoretically that the optima parameters may be equivalently expressed as the solution of a special (convex) Lasso problem. The argument revolves around certain symmetries which are present in their special form of neural network.

**Strengths:**

The search for symmetry has a long and fruitful history in simplifying otherwise intractable problems. Turning the same lens to neural network learning is a natural and possibly fruitful, if challenging, endeavour. Similarly, finding novel methods of solving the neural network optimisation problem could lead to new efficient learning algorithms. In this respect, the authors' contributions appear to be novel.

**Weaknesses:**

1. I am left wondering whether the 'narrow deep network' (equation (3)), which is the focus of the authors' results, can really offer insights which generalise to standard neural networks. This network has a very specific structure in its weights, and I feel it is slightly misleading of the authors not to comment on this. In particular, the weights appear to be structured such that each (scalar) neuron takes its input from just a single neuron in the previous layer, and feeds its output to just a single neuron in the subsequent layer. In actuality, therefore, this 'network' is more like a 'chain'! (Well, several chains added together, but our focus is on the *nonlinear* part of the model, not on taking linear combinations of models.) Can the authors elaborate on why this particular form of network is required to prove the results? Are analogous results expected to hold for 'standard' networks?
2. The authors mention that their numerical results were computed using a 'standard' network, 'to demonstrate that the Lasso model can be useful for this architecture as well'. It is not clear, however, how the numerical results connect with the theoretical results for the simplified model, aside from the slightly vague presence of symmetrical structures. Aren't these symmetrical structures to be expected from the use of the absolute value activation function anyway? The authors should clarify precisely what connection they are trying to draw between the two families of networks here.

**Questions:**

Please see the Weaknesses box.


The Associate Program Chairs have got in touch with me and specifically requested that I add the following questions to the authors, so I am passing these on here:

"
1. Discuss the limitations of generalizing their results to standard neural networks
2. Explain why this particular network structure was chosen and if it was necessary for proving the results
3. Comment on whether they expect similar results to hold for standard network architectures, and if so, what modifications to the proofs might be needed
4. Provide a more detailed explanation of how the numerical results support or relate to the theoretical findings for the simplified model
5. Clarify what specific symmetrical structures in the results are evidence of the Lasso model's applicability, rather than just artifacts of the absolute value activation
6. Discuss any quantitative metrics or qualitative features in the results that demonstrate the usefulness of the Lasso model for standard architectures
7. Address any limitations in applying the theoretical results to the standard network architecture used in the experiments

"

---

### Official Review · Reviewer_P5vV · 2024-11-03

**Soundness:** 3
**Presentation:** 1
**Contribution:** 1
**Rating:** 3
**Confidence:** 3

**Summary:**

This paper proves that training 3-layer deep **narrow** networks with the absolute value activation can be reformulated as a convex Lasso problem with features expressed using geometric algebra. The features for this Lasso problem reveal that such DNNs favor symmetric structures when learning functions that fit the training data. The paper also provides some insights into deeper networks, proving that as the number of layers increases the complexity of the reflections also grows. Finally, the paper provides numerical experiments on synthetic data and language embeddings to validate the theoretical claims.

**Strengths:**

- Understanding what kinds of features deep neural networks learn is an important and unsolved problem. This paper takes a step towards addressing this. albiet in a simplified setting.
- While the regularization and the architecture are highly unconventional, the results apply for multi-variate neural networks.

**Weaknesses:**

The first concern I have is that, despite the claims in the paper, this does not seem to make the learned features more interpretable. For instance, the explicit characterization of the features in Theorem 4.1 is quite complex and hinges on knowing which $2d-1$ subset of the data is involved in $f_j(\mathbf{x})$. The paper does not describe how this can be deduced before training the model.

The other major concern is that the paper relies on a number of simplifications which only losely represent the DNN training problems for even the simplest architectures (a multi-layer ReLU neural network). It is therefore difficult to determine how much of these insights translate to more realistic DNNs. In particular:
- The neural network training problem considered in (2) is not standard the use of $\ell_1$ regularization on the weights is almost never employed when training deep neural networks.
- The majority of the results apply to 3-layer deep narrow networks, that is, deep narrow networks which appear to have only 1 neuron per layer. While there are some results on deeper networks the features learned are much less explicit.
- The results only apply to networks with the absolute value activation. While this limitation is discussed in the paper it is not clear whether they are indicative of what ReLU DNNs will learn in the multi-variate setting.

I will also note that the abstract does not mention any of these simplifications (besides the absolute value activation). It should be updated to make these limitations explicit.

**Questions:**

- Does Theorem 3.1 apply only to deep narrow networks with absolute value activation?

---

### Official Review · Reviewer_4pw2 · 2024-11-04

**Soundness:** 2
**Presentation:** 2
**Contribution:** 1
**Rating:** 3
**Confidence:** 3

**Summary:**

The paper shows an equivalence between training optimal weights for a feed-forward neural network with an absolute value activation function and solving a Lasso problem from convex optimization theory. The absolute value activation function lends a geometric interpretation involving reflections.

**Strengths:**

The paper reduces training a certain kind of neural network to solving a convex optimization problem. This could have a profound impact an an alternative training approach. The paper also asks an important question: how do deep networks learn features differently from shallow networks?

**Weaknesses:**

The main weakness of the paper is that it doesn't answer the question it poses at the beginning: "Is there a fundamental difference in functions learned by deep vs. shallow networks"? The answer the paper gives, which is that deep nets favor symmetric structures, seems to only be true because the authors select an activation function that is symmetric, f(x) = |x|. Section 4.1 purports to extend to ReLU networks but the only comparison made is for 2-layer networks if one adds skip connections, concluding that extending further is an "area for future analysis."

There are several other areas the paper could improve.
1. The experiments do not seem to show anything unexpected. Figure 5 presents plots of a neural network trained with hidden dimension 1 for visualization. The figure's caption seems to make the argument that the depth of the network is causing multi-level symmetry. But since f(x) = |x| is symmetric and being composed, this symmetry and composition seems to be the explanation rather than anything about it being a neural network. If the authors want to improve this point, I might recommend showing similar plots for ReLU networks. Even still, if the hidden dimension of the network is 1, the network loses almost all expressive power. It would be more convincing if the authors trained with a much higher hidden dimension (e.g., 128) and developed a metric that would evaluate the amount of symmetry and reflections in the network's output. Then the authors could present a table showing that as network depth grows, the symmetry metric grows even though it can no longer be directly visualized.
2. Line 030 says, "Research literature still lacks in intuitively understanding why deep networks are so powerful: what they 'look for' in data, or in other words, how each layer extracts features." In fact, there is a literature on how neural networks extract features (see Large et al. 2024, Scalable Optimization in the Modular Norm for one example). Furthermore there is a line of work showing the expressivity of deep networks vs. wide networks going back to the literature on universal approximators (e.g., Multilayer feedforward networks are universal approximators by Hornik, 1989).
3. Some equations are incorrect, undefined, or unnecessarily complicated. For example, Equation 11 which defines a feature function does not multiply by weights except in the first layer. As written, after the first layer, all that occurs is adding biases and applying f(x) = |x|. Is this what the authors intended? Second, several equations do not fully define terms. For just one example, Equation 13 does not define its use of \sigma, making it unclear whether \sigma is the absolute value activation function, ReLU, or a different function. Third, though this is a stylistic point, writing the neural network equations in one line with ellipses in the absolute value signs makes it confusing to read. More relevantly, the geometric algebra formulas the authors write in Theorems 4.1 and 4.3 will take the reader a long time to digest. It could help if the authors simplified the formulas, maybe through redefining variables to make equations less index-heavy.
4. Typos: "Interpretibility" on line 311, "orignal" on line 499, "github" uncapitalized on line 531. Please fix typos before submitting.

**Questions:**

Could the authors explain whether they expect the same symmetry structure in ReLU or GeLU networks?

Why is geometric algebra the right framework to understand neural network geometry?

Would these symmetry results extend to other architectures such as transformers?

What is the benefit or motivation for showing symmetry results? How does it help the machine learning field?
- One direction you could go is leaning into the convex optimization solution for optimal neural network weights. If you could extend your results to find optimal weights via convex optimization for a more general class of neural networks (resnets, transformers, etc.) it could have profound implications for training.

---

### Note · Authors · 2024-11-27

I have read and agree with the venue's withdrawal policy on behalf of myself and my co-authors.